



# Vertical and horizontal variability and representativeness of the water vapor isotope composition in the lower troposphere: insight from Ultralight Aircraft flights in southern France during summer 2021

Daniele Zannoni[1,2], Hans Christian Steen-Larsen[1], Harald Sodemann[1], Iris Thurnherr[3], Cyrille Flamant[4], Patrick Chazette[5], Julien Totems[5], Martin Werner[6], Myriam Raybaut[7]

[1] Geophysical Institute, University of Bergen, Bergen, Norway
[2] Department of Environmental Sciences, Informatics and Statistics, Ca' Foscari University of Venice, Venice, Italy
[3] Institute for Atmospheric and Climate Science, ETH Zurich, Switzerland
[4] Laboratoire Atmosphères, Milieux, Observations Spatiales (LATMOS), UMR 8190, CNRS–SU–UVSQ, Paris, France
[5] Laboratoire des Sciences du Climat et de l'Environnement (LSCE), UMR 1572, CEA–CNRS–UVSQ, Gif-sur-Yvette, France
[6] Alfred Wegener Institute (AWI), Helmholtz Centre for Polar and Marine Research, Bremerhaven, Germany
[7] DPHY, ONERA, Université Paris-Saclay, Palaiseau, France

*Correspondence to*: Daniele Zannoni (daniele.zannoni@uib.no)

**Abstract.** The isotopic composition of water vapor is a valuable tool to track atmospheric hydrological processes and to evaluate numerical models simulating the water cycle. To ensure accurate model-observation comparisons, understanding the spatial and temporal distribution of water vapor isotopes in the troposphere is crucial. The challenging task of obtaining

highly resolved water vapor isotopic observations is typically addressed through airborne measurements performed onboard conventional aircrafts, but these offer limited microscale insights. This study utilizes observations from ultralight aircraft to examine the water vapor isotopic composition in the lower troposphere of southern France during late summer 2021. By combining the observations with conceptual and numerical models, we identify the main processes driving vertical and spatial variability of isotopic composition and we highlight the detection of short-lived, small-scale processes. The key

findings of this study are that (i) at the hourly and sub-daily scales, vertical mixing is the dominant process affecting isotopic variability in the lowermost troposphere and boundary layer above the study site; (ii) evapotranspiration significantly impacts the water vapor isotopic signature, as revealed by the $\delta^{18}$O-$\delta$D relationship; (iii) measurable structures of the water isotopic fields emerge on the scale of 100s of m. The latter are particularly evident for $\delta$D, which also exhibit the largest differences in horizontal and vertical gradients. When combined with other airborne datasets, our results support a simple

model forced with surface observations to simulate the vertical distribution of tropospheric $\delta$D, enhancing the comparison between surface observations and satellite data.





## 1 Introduction

Water vapor is one of the most important gasses driving the dynamics of the Earth's climate system (Fersch et al., 2022, IPCC 2007, Stevens and Bony, 2019). Nearly 99% of atmospheric water vapor resides in the troposphere where it plays a
key role in the formation of clouds and the evapotranspiration process over land and oceans. Stable water isotopes are valuable for studying atmospheric water processes because each phase change impacts the stable isotopic ratio of water at the molecular level. Thus, stable water isotopes provide an essential tool for tracking the hydrological cycle at various spatial and temporal scales (Galewsky et al.: 2016, Dee et al., 2023). In atmospheric water cycle research, the isotopic composition of water vapor is studied alongside the water vapor mixing ratio ($H_2O$, ppmv) or specific humidity (q, g kg$^{-1}$)
because different processes delineate distinct patterns in the $\delta$-humidity space. Here the $\delta$-notation expresses a relative deviation of the stable isotope ratio of a water (vapor) sample from a common reference standard in permille unit (‰) as follows:

$$\delta = \frac{R}{R_{Standard}} - 1 \ (1)$$

where R is the isotopic ratio of heavy to light isotopes of hydrogen (D/H for $\delta$D) and oxygen ($^{18}O/^{16}O$ for $\delta^{18}O$),
respectively, and the "Standard" subscript denotes the ratio in the international standard V-SMOW (Gat, 1996). For instance, in this notation, the turbulent mixing of two air parcels with different mixing ratios and different isotopic composition is outlined by a hyperbolic shape in q, $\delta$ space, while distillation occurring during air parcel drying forms a logarithmic curve (Kendall and McDonnell, 1998; Noone 2012). A commonly used second-order parameter linked to the $\delta$D and $\delta^{18}O$ isotopic composition of water is deuterium excess (d-excess = $\delta$D $-$ 8*$\delta^{18}O$), which provides additional information on non-
equilibrium isotopic fractionation processes. Such processes, like evaporation from a water surface, from water droplets, or condensation of ice crystals are more sensitive to the humidity gradient giving rise to a deuterium excess signature (e.g. Bolot et al., 2013; Merlivat and Jouzel 1979; Zannoni et al., 2022).

Weather regimes, surface topography, air parcels source-sink history all influence the water vapor $\delta$D, $\delta^{18}O$ and d-excess at global and regional scales (e.g., Bonne et al., 2015; Dütsch et al., 2018; Smith and Evans, 2007; Steen-Larsen et al. 2015;
Weng et al., 2021). However, uncertainties remain regarding the control of water vapor isotopic composition in the lower troposphere at meso- and microscales (Aemisegger et al., 2015). Specifically, the extent to which water vapor concentration and isotopic composition can resolve different atmospheric processes is still unclear (Graf et al., 2019). Although the number of observations of the isotopic composition of water vapor has significantly increased in the last 10 years (see e.g. Wei et al., 2019), most of the recent water vapor isotope observations are sparse ground-based measurements of dedicated campaigns
(e.g., Aemisegger et al., 2014; Steen-Larsen et al. 2017). Direct vertical observations in the contiguous troposphere are still scarce and challenging to obtain, especially in the boundary layer. This scarcity is indeed a limiting factor when investigating small-scale and short-lived processes of the water vapor isotopic composition. Remote sensing on satellites provided an important breakthrough to this end, providing nearly global coverage of $H_2O$ and HDO pairs at daily and sub-daily resolution (see e.g. Frankenberg et al., 2013; Herbin et al., 2007l; Schneider et al., 2016; Schneider et al., 2020; Worden et al. 2006;





Zadvornykh et al., 2023). However, satellite data still requires validation with dedicated airborne data (Thurnherr et al., 2024).

Airborne observations are a suitable tool to investigate the horizontal and vertical distribution of water stable isotopes in the troposphere. Notable airborne measurements have been performed in the last 10 years, such as for the HyMeX project in the Mediterranean area (Sodemann et al., 2017) or over the subtropical North Atlantic Ocean for the MUSICA project (Dyroff et

al., 2015) and western tropical North Atlantic for the EUREC4A project (Bailey et al. 2022). Recently, both Unmanned Aerial Vehicles (UAV) and Ultralight Aircrafts (ULA), such as ultralight trikes, have been used to observe the isotopic composition of water vapor, complementing conventional propeller-driven aircraft (Chazette et al., 2021, Rozmiarek et al., 2021). Despite challenges from large temperature variability due to the open fuselage pod and strong vibrations from proximity to the aircraft engine, ULAs equipped with Cavity Ring-Down Spectroscopy (CRDS) analyzers can provide

highly resolved spatial and temporal information on water vapor isotope composition over large areas (>20 km²) within the lower troposphere (≤3500 m ASL) multiple times within a day. These characteristics are essential for evaluating both the spatial and temporal representativeness of water vapor isotope composition observations in the troposphere. In this study, we utilize highly temporally and spatially resolved water vapor isotopic observations collected with an ULA during late summer 2021 in a Mediterranean climate region to provide insights into the main driving factors of the variability of water vapor

isotopic composition in the lower troposphere (Zannoni et al., 2023). Specifically, our primary objective is to determine the horizontal and vertical variability of the stable water vapor isotope composition in the boundary layer and in the lowermost free troposphere. We further explore the drivers of the spatial short-lived and small-scale water isotope pattern using conceptual and numerical models and assess to which degree ground-based water isotope observations provide information about the vertical water vapor isotope structure.

## 2 Materials and methods

### 2.1 Study site and flight overview

From 17 Sep 2021 to 23 Sep 2021, 13 flights were performed with an ULA near Aubenas (southern France) to probe the vertical and spatial structure of the isotopic composition of water vapor in the boundary layer and lowermost free troposphere (Table 1 and Fig. 1). Takeoff, landing and ground operations were conducted next to the Aubenas Aerodrome

(ICAO: LFHO). LFHO is located on the top of a plateau bordering the west side of the Rhône Valley. The area is surrounded by low altitude hills and mountains and is characterized by a Mediterranean climate. During the study period, the minimum and the maximum temperatures were 16 and 30 ˚C, respectively. Even though convective thunderstorms passed the area, only a single low-intensity precipitation event was recorded at the site during the night between 18 and 19 Sep 2021. Wind conditions only prevented flight operations on 19 Sep 2021 afternoon, when southerly winds of up to 14 m/s prevailed.



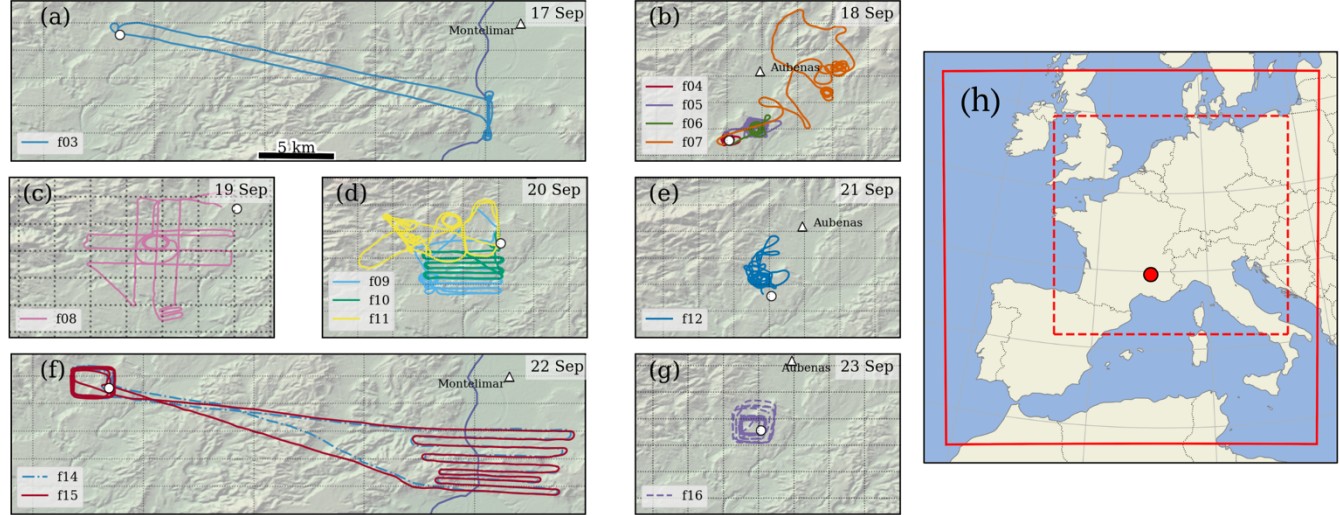

**Figure 1: ULA flights f3 to f16 over the area of Aubenas (Aubenas Aerodrome) on each flying day in September 2021 (a-g). The airfield area is depicted in all the panels as a white circle. The towns of Aubenas and Montelimar are reported for reference as white triangles. The Rhône Valley is visible on the east side of the map in panels a and f. Horizontal scale reported in panel a (5 km) is valid for panels a-f. (h) Geographical location of the Aubenas Aerodrome in France and COSMO$_{iso}$ domains for coarse (0.1˚x0.1˚, solid red) and fine (0.02˚x0.02˚) resolutions.**



**Table 1**: Overview of the flights performed between 17 Sep 2021 and 23 Sep 2021. Time in Coordinated Universal Time (UTC). Altitude in meters Above Mean Sea Level (m ASL).

| Flight (ID) | Date (dd/mm/yyyy) | Takeoff (HH:MM) | Landing (HH:MM) | Max altitude (m ASL) | Objective |
|---|---|---|---|---|---|
| f03 | 17/09/2021 | 15:28 | 16:47 | 3100 | Test flight toward the Rhône Valley |
| f04 | 18/09/2021 | 05:12 | 06:06 | 1669 | Diurnal profile, early morning flight |
| f05 | 18/09/2021 | 08:16 | 09:25 | 1730 | Diurnal profile, morning flight |
| f06 | 18/09/2021 | 12:16 | 13:09 | 1751 | Diurnal profile, midday flight |
| f07 | 18/09/2021 | 14:55 | 16:05 | 3157 | Diurnal profile, afternoon flight |
| f08 | 19/09/2021 | 07:57 | 09:29 | 2166 | Vertical profile and spatial scan covering ~10 km x 10 km area |
| f09 | 20/09/2021 | 06:42 | 08:28 | 2162 | Spatial sampling: 600, 1200 m ASL |
| f10 | 20/09/2021 | 09:37 | 10:53 | 1254 | Spatial sampling: 700, 900, 1200 m ASL |
| f11 | 20/09/2021 | 16:04 | 17:46 | 3120 | Sampling below and above clouds |
| f12 | 21/09/2021 | 06:57 | 08:37 | 3173 | High altitude profile |
| f14 | 22/09/2021 | 08:00 | 09:55 | 3141 | Scan of Rhône Valley and vertical profile |
| f15 | 22/09/2021 | 13:00 | 15:07 | 3204 | Scan of Rhône Valley and vertical profile |
| f16 | 23/09/2021 | 08:04 | 09:47 | 3163 | High altitude vertical profile, highly resolved pattern below 1500 m ASL |



## 2.2 Water vapor isotopic composition measurements

A Tanarg 912 XS ULA (Air Création, flown by Tignes Air Experience) was equipped with a CRDS water vapor isotope analyzer from Picarro (model L2130-i, s/n HIDS2254, hereafter CRDS analyzer). The CRDS analyzer is the same that has been used in Chazette et al. (2021) and was placed on the back seat of the ULA. To minimize the effect of the large ambient temperature variability on the CRDS analyzer performances, the analyzer was wrapped with a layer of 3 mm thick neoprene sheet (RS 733-6757). A foldable aperture was made on the wrapping sheet to ensure air ventilation on the backside of the instrument. Ambient air was sampled by the CRDS analyzer in flight mode at a nominal flow rate of 80 sccm min$^{-1}$ through an unheated inlet of 80 cm length (1/4-inch O.D. stainless steel with Silconert coating) pointing backward on the right side of the aircraft. Despite the lack of inlet heating, no evidence of condensation was observed in the isotope data. This is likely due to the short length of the inlet, resulting in minimal air residence time within the system, as well as the ULA's infrequent exposure to high relative humidity conditions. The CRDS analyzer was set in flight mode, which enabled to measure water vapor volume mixing ratio ($H_2O$, ppmv), $\delta^{18}O$ and $\delta D$ (‰) at ~4 Hz sampling rate, hence more responsive than conventional operating mode (~40 sccm min$^{-1}$, ~1Hz). $H_2O$ (ppmv) was converted to specific humidity q (g kg$^{-1}$) following Vaisala (2023). For both VSMOW-SLAP and humidity-isotope dependency calibration, the inlet was connected with a 3-way valve to a water vapor generation module that allowed the injection of water isotope standards for q ranging between 0.6 and 12 g kg$^{-1}$ (Steen Larsen and Zannoni, 2024). Three water isotope standards provided by FARLAB, University of Bergen, were used every day, bracketing all the potential isotopic variability in water vapor isotopic composition in the lower troposphere of the study area (details on frequency of usage and values reported in Supplementary Material SM0). The VSMOW-SLAP slope of the calibration line varied between 1.118 - 1.132 and 0.914 - 0.928 for $\delta^{18}O$ and $\delta D$, respectively, with no visible trend during the study period. Such slope values are consistent with the long-term slope variability of the instrument estimated between 2016-2022 ($\delta^{18}O$ slope = 1.1305 ± 0.0095, $\delta D$ slope = 0.9253 ± 0.0027), thus ensuring reliable instrument performances during the field operations. Four characterization curves were performed to check the consistency of the humidity-isotope dependency between laboratory test and field deployment (not reported). Calibration of q was performed once in the range 1.2 - 12 g kg$^{-1}$ using a calibrated chilled mirror hygrometer (Panametrics OptiSonde) as the reference instrument. The dry air source was obtained with a dry air compressor from (cleanAIR CLR 20/25) equipped with an extra drying cartridge in series (Agilent MT400-4). The humidity level of the provided dry air was <0.06 g kg$^{-1}$.

## 2.3 Precision and accuracy of water vapor isotope observations

A 90-minutes injection of BERM standard on 22 Sep was used to investigate the instrument precision in stable condition on the field with the ULA engine turned off. The first 30 minutes of the injection were discarded, to ensure an acceptable removal of the memory effect in the inlet. The remaining 60 minutes were used to run an Allan deviation (ADEV) test at q =8.3 ± 0.3 g kg$^{-1}$, yielding 0.25 second ADEV of 0.20‰, 0.74‰ and 1.87‰ for $\delta^{18}O$, $\delta D$ and d-excess, respectively and 1



second ADEV of 0.10‰, 0.38‰ and 0.95‰ for $\delta^{18}$O, $\delta$D and d-excess, respectively (for figure, see Supplementary Material SM1), typical of L2130-i series. However, these values cannot be used as a reference for the precision of the instrument in flight conditions. Given that the L2130-i model uses peak absorption height for the spectral fitting, the precision of the

instrument is highly sensitive to pressure broadening caused by vibrational noise transmitted by the ULA engine. As an example, Supplementary Material SM2 shows how cavity pressure, $\delta^{18}$O and d-excess measurement noise increase when the ULA engine was turned on just before takeoff for flights 7, 8, 9. Assuming that the isotopic composition of atmospheric water vapor did not change significantly 30 seconds before and 30 second after turning on the engine, the standard deviations of $\delta^{18}$O, $\delta$D and d-excess calculated over 1 minute provide insights on the decrease of instrumental precision due to engine

vibrations. The standard deviations with engine off (on) resulted 0.22 (0.45) ‰, 0.78 (0.99) ‰ and 1.92 (3.54) ‰ for $\delta^{18}$O, $\delta$D and d-excess, respectively, at q = 8.2 ± 0.4 g kg$^{-1}$. Assuming white noise for averaging time between 0.25 and 10 seconds, it is possible to normalize the results of the ADEV for when the engine is running, yielding 1 second ADEV of 0.23‰, 0.50‰ and 1.78‰ for $\delta^{18}$O, $\delta$D and d-excess, respectively. These ADEV values can therefore be assumed representative of the instrumental precision at 1 second averaging time and at q = 8.2 g kg$^{-1}$. Note that shocks and vibrations

are expected to be less pronounced when the ULA is airborne, thus we provide here a conservative estimate of the vibrational impacts.

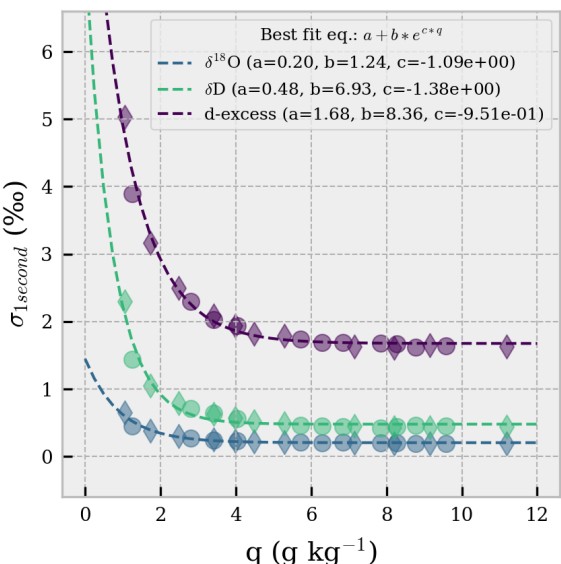

**Figure 2: Precision of the CRDS analyzer as a function of humidity affected by ULA engine vibrations at ground level. Circles and diamonds represent data from GLW humidity-isotope characterization performed on 19 and 20 September, respectively. Dashed**
**lines are best fit curves.**

Similarly, the 0.25 seconds standard deviations for $\delta^{18}$O, $\delta$D and d-excess measured during each step of the humidity-isotope characterization curves were scaled for averaging time of 1 second and accounting for engine vibrations (Fig. 2).



Instrumental precision can therefore be considered constant between 4 - 12 g kg$^{-1}$, with a rapid decrease at low humidity ($\sigma_1$ second is 0.7‰, 2.9‰ and 8.0‰ at q = 1 g kg$^{-1}$ for $\delta^{18}$O, $\delta$D and d-excess, respectively).

**2.4 Postprocessing of the water vapor isotopic composition signal**

The measuring system of the isotopic composition of water vapor is characterized by its own response time, which in turn depends on the inlet design as well as on the characteristics of the CRDS analyzer itself (Aemisegger et al., 2012, Steen-Larsen et al. 2014). When working with high frequency data such as for airborne measurements, it becomes important to consider the response time of the measuring system. Indeed, different response times for q, $\delta^{18}$O, $\delta$D can introduce artifacts when looking at a combination of the signals (e.g q vs isotopes, or $\delta^{18}$O vs $\delta$D for d-excess). The impulse response of the system was estimated by inducing a large humidity and an isotope step change and by performing the spectral analysis of its first derivative. Briefly, using a 3-way valve operated by the CRDS analyzer software, the inlet source was switched between ambient air and dry air, for humidity analysis, and between ambient air and standard water vapor for isotope analysis at the same humidity level (Fig. 3.a). The test was repeated three times. The raw data of the CRDS analyzer was studied at the sampling frequency of the analyzer (4 Hz) to avoid any possible artifacts introduced by applying a running average or by data resampling.

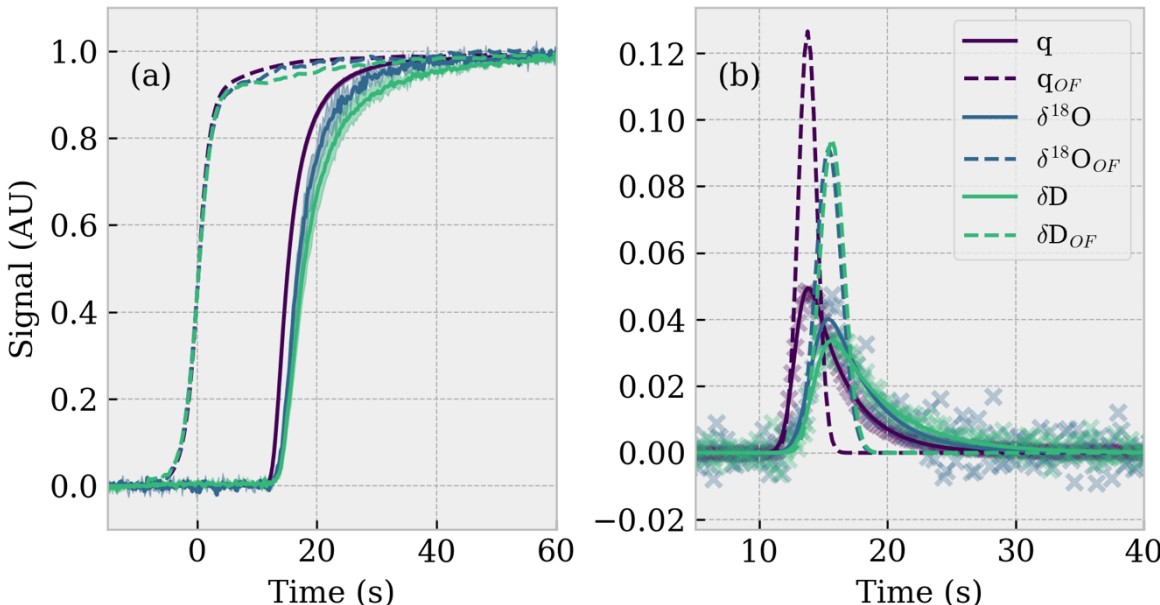

**Figure 3: Analysis of the response of the CRDS analyzer to a Heaviside step-function in q and in change in isotopic composition. (a) Min and max normalized step change (arbitrary units, AU) for q, $\delta^{18}$O and $\delta$D (averaged over 3 repetitions). Solid lines and shadings are average ± 1 standard deviation of raw observations of the three repetitions, respectively. Dashed lines represent filtered and sync data. Origin of the horizontal axis set when the 3-way valve was switched from ambient air to the calibration line. (b) Exponentially Modified Gaussian (EMG) best fit of the 1$^{st}$ derivative of the observed step changes (solid lines). Gaussian impulses with the same areas of EMG impulses (dashed lines).**



First, the delay introduced by the inlet + analyzer was estimated by measuring the time required to observe a deviation of the signal larger than 2σ when compared to the previous average state. Such delay was estimated to be 13.75 ± 0.05, 15.36 ±

0.27 and 15.60 ± 0.13 seconds for q, $\delta^{18}$O and $\delta$D, respectively. Second, the first derivative of the normalized step change was fitted with an Exponentially Modified Gaussian (EMG) distribution to perform the Fast Fourier Transform and to investigate the impulse response of the system (Fig. 3.b). The result of the fit shows that peaks for q, $\delta^{18}$O and $\delta$D are not symmetrical. In analogy with chromatography (Kalambet et al., 2011), the EMG can explain the peak shape by the convolution of two distinct physical processes: mixing (Gaussian) and absorption/desorption of tubing and cavity walls

(exponential). In this context, the EMG peaks were transformed into the "desired" gaussian peaks by maintaining the same gaussian σ, estimated with EMG fit, and the same area under the peak. An optimal filter (OF) was then designed by calculating the ratio of the transfer functions of EMG and gaussian peak and by applying a 1$^{st}$ order Butterworth low pass filter to remove ringing (frequency cut off 0.1 Hz). The effect of optimal filtering and synchronization of rising edges is reported as dashed lines for q, $\delta^{18}$O, $\delta$D in Fig. 3.a.


## 2.5 Meteorological observations and position data

The ULA was equipped with a fast-response temperature (T, ˚C) and humidity (RH, %) probe iMet XQ-2 (InterMet systems, s/n. 61124) which also provided air pressure (P, hPa) and GPS position at 1 Hz rate. The probe was installed below the wing on the mast of the ULA, ensuring excellent ventilation in flight condition and easy access for maintenance on the ground.

After the postprocessing of q, $\delta^{18}$O and $\delta$D signals as described in section 2.4, no further adjustment was necessary to align the CRDS q time series with the iMet humidity data. For position data, synchronization between Picarro and iMet was achieved by using atmospheric pressure readings, since this specific CRDS model had an atmospheric pressure transducer installed inside the chassis. The pressure readings were used for synchronization between GPS and CRDS instead of humidity readings because of the extremely short response-time of the pressure sensors (in the order of a few tens of

milliseconds).

Several other meteorological parameters were acquired from ERA5 reanalyses, available on the Copernicus Climate Data Store (CDS) (Hersbach et al., 2023). Boundary layer height (blh, m), dew point temperature (d2m, K), surface pressure (sp, Pa) were retrieved from ERA5 hourly data on single levels. For data on single levels, the reanalysis data was interpolated to the Aubenas Aerodrome coordinates. More specifically, the blh variable was adjusted accounting for geopotential (z, m$^2$ s$^{-2}$)

to allow comparison with flight altitude (m ASL). Air temperature (t, K) and specific humidity (q, kg/kg) data was also retrieved as hourly data on pressure levels (37 levels).

## 2.6 Spatial correlation and spatial representativeness of the data

The spatial structure of the water vapor mixing ratio, and its isotopic composition is investigated by means of the variogram

and of the Moran's I spatial autocorrelation index. The variogram is a tool used to describe the variability (semivariance)



between pairs of data points that are separated by a certain lag distance in the 3D space. If a spatial structure exists in the data, the observed semivariance can be explained by means of a statistical model (experimental variogram) and the variable of interest can be predicted in-between non-observed locations. The experimental variogram usually starts from a non-zero value (the *nugget* term) and increases until reaching a plateau (the *sill* term) within a certain distance (the *range* term, set at

95% of the sill). Using such terminology, the range can be understood as the maximum distance at which observations are correlated. Several models can be used to fit the observed semivariance, in this study we used the *spherical model*, which is the standard choice when fitting the empirical variogram using the Python package SciKit-GStat (Mälike, 2022). The Moran's I, on the other hand, is a statistical test to measure the degree of spatial autocorrelation (also reported as the Global Moran's I, ESRI 2024). Its null hypothesis is that the variable under investigation is randomly distributed in the study region.

Hence, similarly to the Pearson correlation index, the Moran's I ranges between -1 and 1, where -1 indicates that observations tend to be dispersed and 1 indicates the tendency of observations toward clustering. A Moran's I value close to 0 indicates the absence of spatial autocorrelation. The Python package PySAL has been used to estimate Moran's I by attributing spatial weights with the *distance band* method (Rey and Anselin, 2007).

**2.7 Conceptual models describing the vertical profile water vapor isotopic composition**

To simulate the vertical profile of water vapor isotopic composition two conceptual models were used: a Rayleigh distillation model and a binary mixing model. Both conceptual models are widely used for describing and generalize the variability of the isotopic composition of atmospheric water vapor. The reader is referred to the literature for a full description of their validity and their mathematical derivation (Galewsky et al., 2016; Gat, 1996; Noone, 2012 and references

therein). Specifically, here we report only the principal assumptions behind the two approaches, and we refer to equations in Noone (2012) for both models.

In the Rayleigh model the decrease in air temperature due to adiabatic lift in saturated conditions (RH=100%) drives the reduction of the saturation vapor pressure of the air. Under the assumption that excess water is completely removed immediately after the phase change, the isotopic ratio of the remaining water vapor follows a logarithmic curve whose shape

is given by the temperature-dependent equilibrium fractionation factor between vapor and liquid or vapor and ice (eq. 12 as seen in Noone 2012). The average of the observations collected with the ULA at the lowest model level for each flight were used as the initial conditions for the Rayleigh model.

In the binary mixing model, the only process involved is the turbulent mixing between two *end members*: dry air coming from the free atmosphere and the water vapor flux from the surface (evapotranspiration). The main point of this model is that

no isotopic fractionation is involved in the process. Mixing will make humidity and isotopic composition tend toward a well-mixed state with a hyperbolic curve connecting those two extreme values. An important assumption in this model is that vertical mixing between layers is the only active process. The average of the observations collected with the ULA at the highest level available for each flight was used as representative of the dry end member ($q_0$ and $\delta_0$ as seen in Noone, 2012, eq. 23). A linear fit between the upper (drier) end member and the average of the observations at the lowest level (moist) was




used to identify the flux composition ($\delta_F$ as seen in Noone, 2012, eq. 23). Finally, for each flight and for both models the atmospheric column above the study area was discretized into 20 evenly spaced layers, from 300 to 3300 m with a 150 m constant layer height.

## 2.8 COSMO$_{iso}$ simulations

In addition to conceptual models, the isotope-enabled regional weather prediction model COSMO$_{iso}$ (Pfahl et al., 2012) was used to investigate the vertical and spatial structure of the isotopic composition of water vapor. Two additional water cycles for the heavy water molecules $H_2^{18}O$ and $HD^{16}O$, respectively, are implemented in COSMO$_{iso}$ to simulate the isotopic composition of the atmospheric water cycle. The additional water cycles behave analogously to the $H_2^{16}O$ water cycle and, additionally, include isotopic fractionation during phase change processes. A 10-day COSMO$_{iso}$ simulation from 15 to 24

Sep 2021 at 0.1° (~10 km) horizontal resolution and a 5-day simulation from 16 to 21 Sep 2021 at 0.02° resolution (~2 km) have been conducted. The domain of the coarser simulation is centered around Aubenas and covers Western Europe including the Mediterranean and Baltic Seas, and the Western Atlantic eastwards of approximately -14°E (Fig. 1 h). The 2km COSMO$_{iso}$ domain lies within the 10 km domain covering France and adjacent coastal ocean basins. The simulations were performed with 41 vertical levels and coupled to the isotope-enabled land module TERRA$_{iso}$ including prognostic

isotopic compositions of terrestrial water reservoirs (Dütsch, 2016; Christner et al., 2018). 6-hourly outputs from the global, isotope-enabled atmosphere model ECHAM6-wiso (Cauquoin & Werner, 2021) provided the initial and boundary conditions. The ECHAM6-wiso wind fields were spectrally nudged to the COSMO$_{iso}$ simulations above 850hPa to ensure a good representation of the large-scale flow in the regional simulations. The global ECHAM6-wiso simulation was conducted at a horizontal resolution of 0.9°, with 95 vertical levels and was spectrally nudged to ERA5 reanalysis data (Hersbach et al.,

270    2020).

The representation of convection in numerical simulations depends on the grid scale and chosen parametrizations. At a horizontal resolution on the order of 10km or less, COSMO (Steppeler et al., 2003) simulations with explicitly resolved convection resulted in a better representation of precipitation distribution over Europe than simulations with parameterised convection (Vergara-Temprado et al., 2019). Further, COSMO$_{iso}$ simulations with and without convection parametrization

showed a good agreement in the isotopic composition of water vapor with satellite observations over West Africa (de Vries et al. 2022). We therefore performed both COSMO$_{iso}$ simulations with explicit convection in accordance with previous studies (e.g. Villiger et al. 2023, Thurnherr et al. 2024).

## 3 Results

### 3.1 Weather situation during the campaign

The overall weather situation during the campaign period can roughly be divided into three phases. During a first phase from 15 to 18 Sep, south-eastern France was in between the influence of North Atlantic air masses belonging to a frontal system




west of the British Isles, and a high-pressure area east of Portugal (Fig. 5a). This period was characterized by low winds and
generally low cloudiness (Fig. 4c), and a large diurnal temperature amplitude with up to 25 ˚C daily maximum temperatures
(Fig. 4b). On 18 Sep, the frontal band had broken apart, shedding a short-wave trough over the Gulf of Biscay, which was
then associated with intense showers over southern France during the night from 18 to 19 Sep (Fig. 4c). This precipitation
initiated the second phase, lasting from 19-20 Sep (Fig. 5b). Inflowing North Atlantic air led to overall cooler temperatures
with daily maxima of 20 ˚C, characterized by more overcast and rainy periods (Fig. 4b, c). The phase ended after an intense
convective rainfall event during the mid-day of 20 Sep. Thereafter, a strengthening of the anticyclone over the Azores
extending towards the English Channel (Fig. 5c, d) led to a mostly cloud-free period with increasing diurnal temperature
amplitudes of up to 12 ˚C (Fig. 4b). Wind gusts reached up to 15 m s$^{-1}$ on 21 Sept., slowly decreasing over the next days
until 24 Sept (Fig. 4d). The ERA5 boundary layer height shows clear diurnal cycles, reaching typically 1000-2000 m above
ground (Fig. 4e).

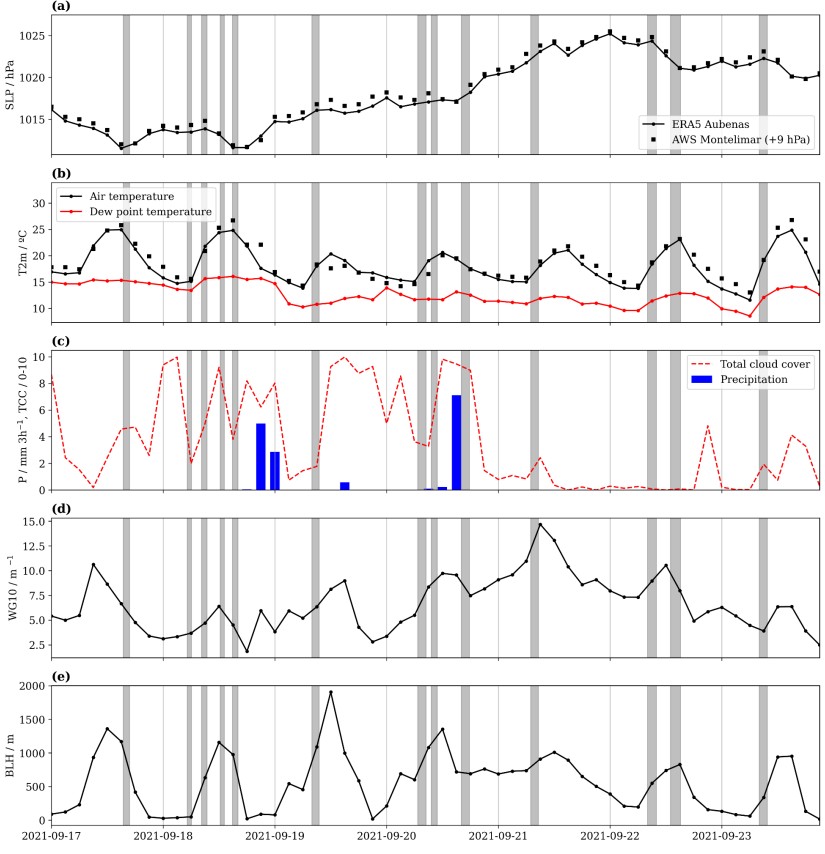

**Figure 4: Evolution of weather parameters from ERA5 at the grid point closest to Aubenas compared to an automatic weather**
**station in Montelimar (ca. 20 km distance in the Rhone valley). Grey shading indicates flight periods. (a) pressure at mean sea**
**level from ERA5 (hPa, dots) and AWS (hPa, squares), (b) air temperature at 2m (ºC, black) and dew point temperature at 2m (ºC,**
**red) from ERA5 and from AWS (ºC, squares), (c) surface precipitation (mm 3h$^{-1}$, bars) and total cloud cover (1/10s, red dashed**
**line), wind gusts at 10 m (m s$^{-1}$), (e) atmospheric boundary layer height (m). Note that an offset of 9 hPa was added to the AWS**
**MSL at Montelimar for easier comparison.**


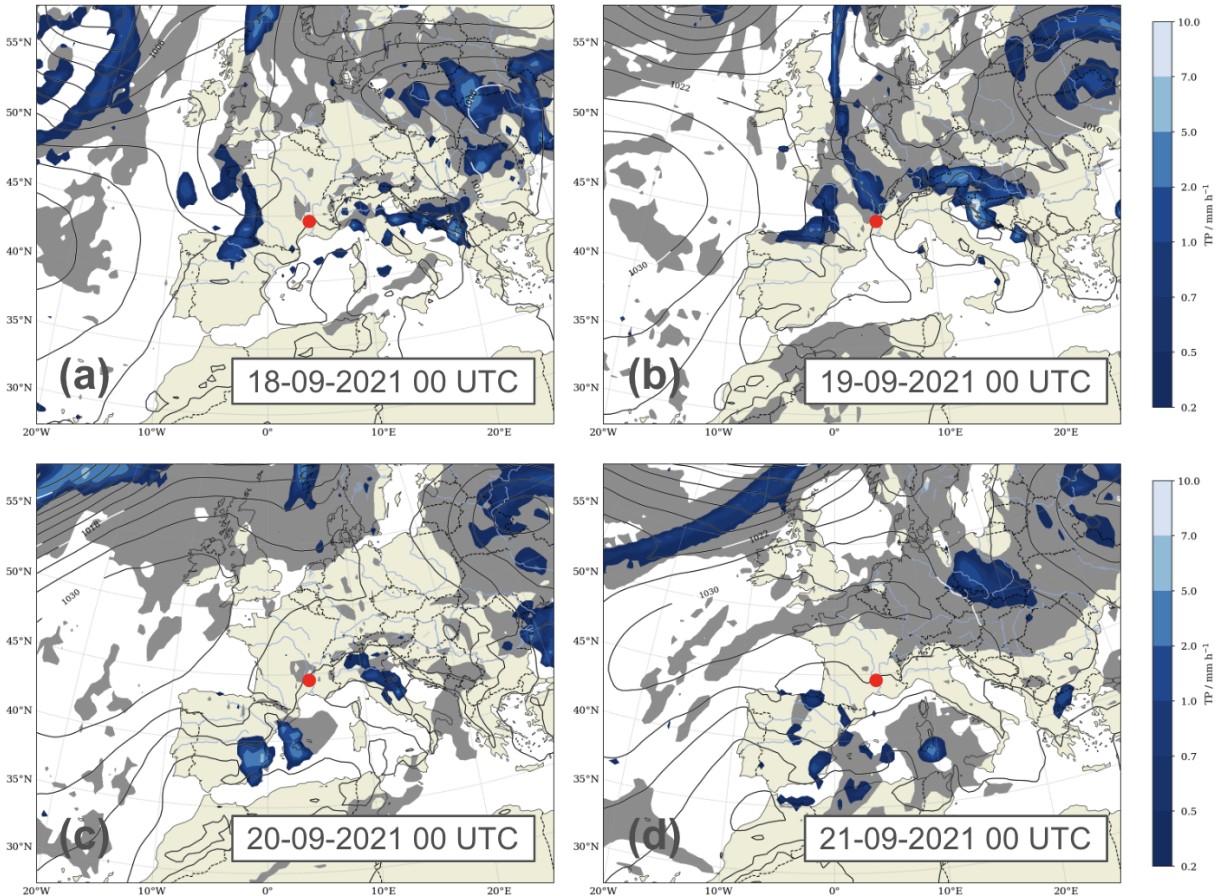

**Figure 5: Total precipitation (blue shading, mm h⁻¹), total cloud cover (gray shading, 0.9 and above), and sea-level pressure (contour interval 4 hPa) from ERA5 at (a) 00 UTC on 18 Sep 2021, (b) 00 UTC on 20 Sep 2021, (c) 00 UTC on 21 Sep 2021, and (d) 00 UTC on 22 Sep 2021.**

**3.2 Daily vertical profiles of the water vapor isotopic composition**

We now investigate the time evolution of the vertical profile measurements from the ULA during the campaign period. Figure 6 shows 150 m binned vertical profiles of potential temperature, specific humidity and water vapor isotopic composition ($\delta D$ and d-excess). $\delta^{18}O$ is not reported in Fig. 6 but is discussed in the text. The potential temperature profiles depict a stable atmosphere for most of the flights above ~1200 m. The binned values of specific humidity and isotopic

composition, fall within a range of [1.1 ; 9.3] g kg⁻¹, [-40.91 ; -15.79 ] ‰, [-315.59 ; -114.25] ‰ and [9.1 ; 19.1] ‰ for q, $\delta^{18}O$, $\delta D$ and d-excess, respectively. The general decrease of mixing ratio and $\delta D$ as a function of altitude is clearly visible. However, the specific humidity decrease with height is rather uniform and mirroring the general potential temperature increase up to 3000 m (for air temperature see panel e in Supplementary Material SM3).



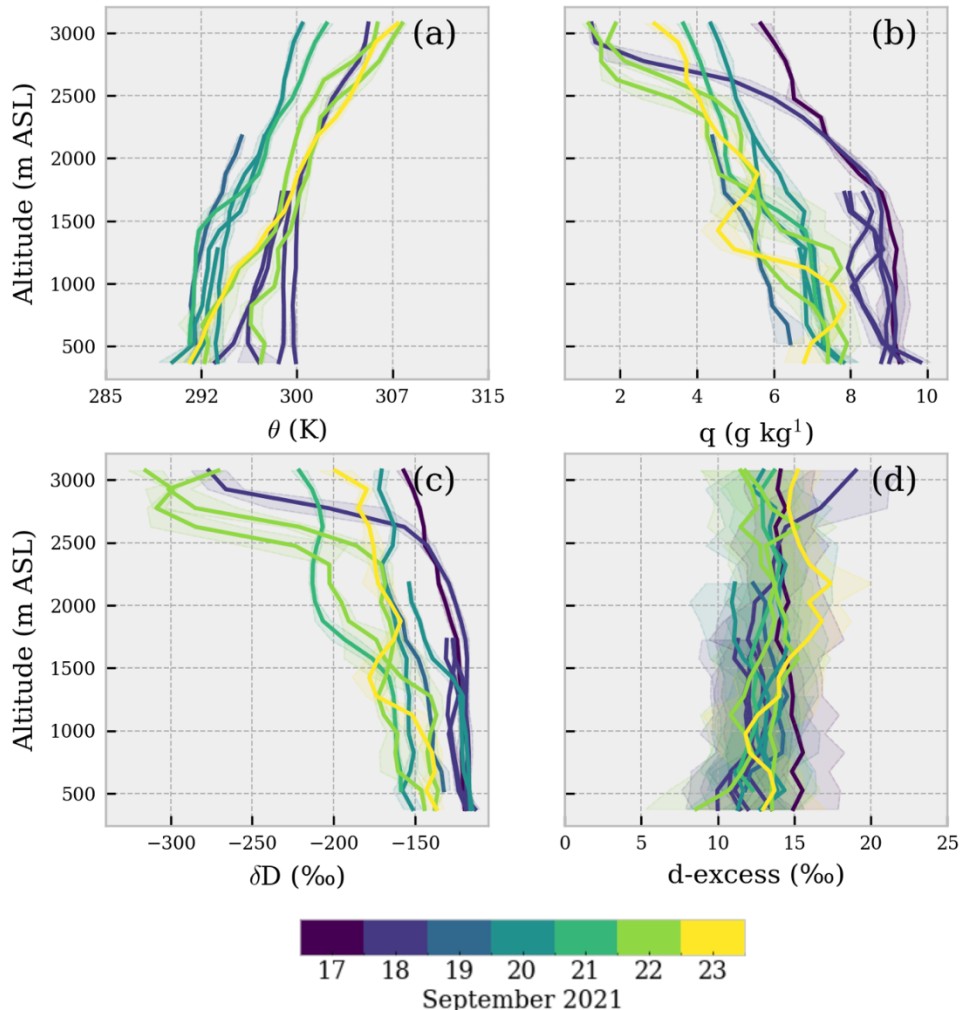

**Figure 6: Vertical profiles of potential temperature (a), specific humidity (b), water vapor δD (c) and d-excess (d). Solid line represents the average calculated over a 150m bin size. Shadings represent ±1σ interval around the mean.**

A pronounced change in $\delta$D is visible at ~2500 m altitude. Using 2500 m as a cutoff altitude, it is possible to define the isotopic lapse rate for $\delta^{18}$O and $\delta$D, which yields -0.20 ± 0.14 ‰ 100 m$^{-1}$ and -1.5 ± 1.2 ‰ 100 m$^{-1}$. These isotopic lapse rates are fully comparable to vertical gradients observed for surface precipitation as a function of the altitude of several sampling stations in the Mediterranean region (see e.g. Balagizi and Liotta, 2019; Masiol et al., 2021).

Below 2500 m, d-excess shows no particular feature for all the flights despite the large RH variability observed (panel f in Supplementary Material SM3). Among the flights which reached altitudes > 3000 m (flights 3, 7, 11-16), only flight 7 exhibits a consistent positive deviation of d-excess from the mean value observed at lower altitudes, ranging from 12 ± 2 ‰ at 2000 m to 19 ± 3‰ at 3000m. The d-excess increase as a function of the altitude is a well-known feature of atmospheric





water vapor and typical of clear sky conditions. Notably, the d-excess increase of flight 7 starts after reaching a maximum of RH centred around 1800 - 2000 m which might be representative of the cloud base level.

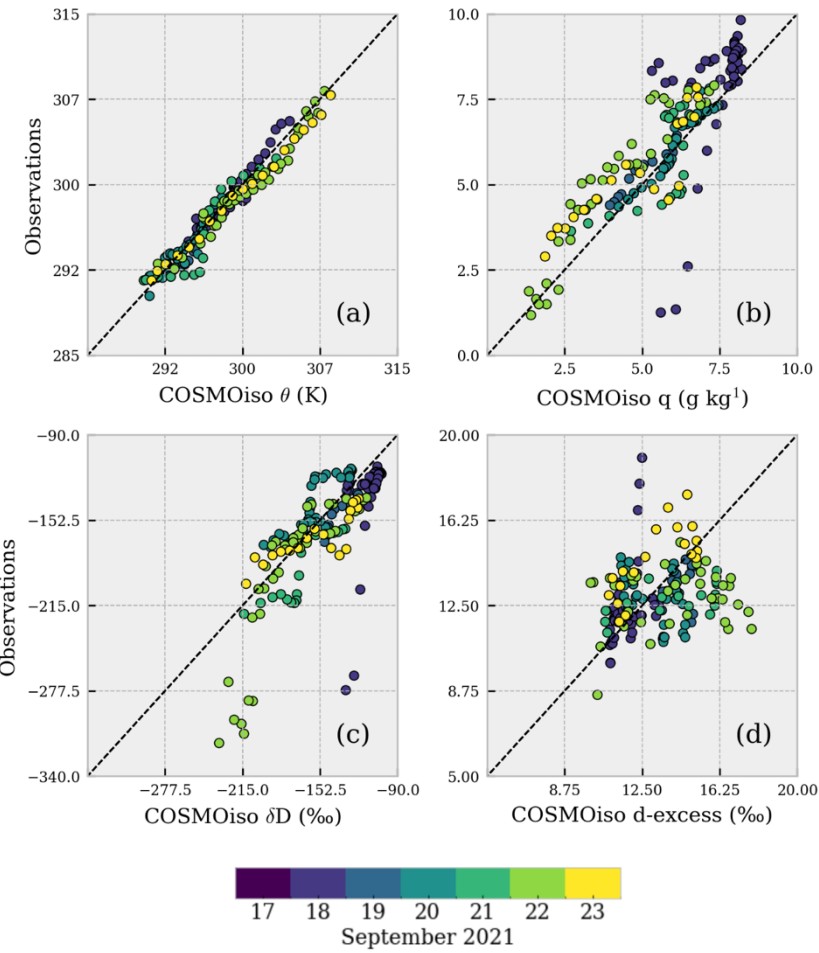

**Figure 7: Comparison between COSMOiso interpolated profiles and observations for the same variables of Fig. 6 (a, b, c, d).**
**Dashed line represents a 1:1 relationship.**

On a temporal perspective, temperature profiles observed on 17 and 18 Sep are similar to profiles observed on 22 and 23 Sep but different to profiles observed on 19-21 Sep. The average lapse rate observed is 6.54 ˚C km$^{-1}$, with min-max ranging 4.10-8.88 ˚C km$^{-1}$, respectively. The temperature variability is characterized by a symmetrical fluctuation of the mean values during the study period. No such fluctuation is observed for specific humidity and water vapor $\delta D$ ($\delta^{18}O$). The fact that humidity and water vapor isotopic composition show instead a monotonic decrease during the campaign likely reflects a large-scale circulation control on the moisture properties.

Potential temperature, q and $\delta D$ simulated by COSMO$_{iso}$ are in close agreement with observations for most of the flights as shown in Fig. 7 (r>0.95 for 7 out of 12 flights, Fig. 6.e-g). Noticeable differences between model and observations are



visible for flights on 18 and 22 Sep (blue and orange circles). The difference in $\delta$ values for 18 Sep flight can likely be attributed to the mismatch in simulated humidity: the COSMO$_{iso}$ model simulates a more humid vertical profile above 2000 m, both in terms of specific and relative humidity, which yields a more enriched water vapor in $\delta^{18}O$ and $\delta D$ at high altitude levels. On the other hand, the difference in $\delta$ values for 22 Sep is not related to differences between simulated and observed humidity profiles. In general, COSMO$_{iso}$ simulates a less depleted water vapor above 2500 m ASL for flights 7, 14, 15,

which are the flights where the largest $\delta^{18}O$ and $\delta D$ gradients was observed (such a bias is on average $10 \pm 5$ ‰ and $80 \pm 37$ ‰ for $\delta^{18}O$ and $\delta D$, respectively). For the d-excess, the COSMO$_{iso}$ model shows a similar or slightly higher variability than the observations which are relatively constant with height. A medium correlation (r >0.5, p-value < 0.01) was found between COSMO$_{iso}$ and observed d-excess profiles for ~50% of the flights but is also worth noting that the direction of the correlation is negative for 3 out of 12 flights (5, 9, 10). Discrepancies between observed and modelled d-excess can be attributed to

differences in simulated and observed $\delta^{18}O$ and $\delta D$ at high altitude, to a weak correlation between observed and modelled RH profiles (r = 0.40) and to the influence of the land surface scheme and how this treats fractionation (Aemisegger et al., 2015).

## 3.3 Water vapor $\delta^{18}O$ vs $\delta D$ relationship in the lower troposphere

All the ULA flights crossed the boundary layer top (blh min, mean, max: 949, 1221, 1681 m ASL, respectively). The observed water vapor isotopic composition retrieved from the ULA can therefore be considered as representative of the water vapor within the boundary layer and can also provide insights about the water vapor composition of the lowest part of the free troposphere. When the $\delta^{18}O$ and $\delta D$ data points from all the flights are combined together, the regression becomes $\delta D = (7.88 \pm 0.003) *\delta^{18}O+(10.53 \pm 0.07$ ‰) (Fig. 7). This regression line matches closely to the Global Meteoric Water

Line $\delta D = 8*\delta^{18}O+10$‰ (e.g., Rozanski et al., 1993). A similar meteoric water line of $\delta D = (7.76 \pm 0.005) *\delta^{18}O+(8.12 \pm 0.09$ ‰) is obtained with COSMO$_{iso}$ interpolated data. A slope close to 8 suggests that the same main process is modulating the water vapor isotopic composition and the isotopic composition of global precipitation. However, the $\delta^{18}O$ vs $\delta D$ slope for each flight ranges from 3.82 to 8.06 indicating that a simple distillation is not the sole process involved. Figure 8 inset indeed depicts an evident positive correlation (r = 0.84, p-value<0.01) between the maximum altitude reached by the ULA

and the $\delta^{18}O$ vs $\delta D$ slope. Such a positive correlation might indicate the imprint of a local evapotranspiration signal in the boundary layer moisture. The blh was then used as a threshold, assuming water vapor being more influenced by the surface evaporation flux below the blh. Table 2 reports evident differences between the $\delta^{18}O$ vs $\delta D$ slopes calculated only within the boundary layer or for the full vertical extent of the flight. A slope value >7 is always observed when the water vapor sampled below the blh accounts for ⪅ 50% of the flight observations, indicating that a $\delta^{18}O$ vs $\delta D$ slope smaller than ~7 is typical of

water vapor sampled within the boundary layer, as observed in several ground based studies (e.g. Aemisegger et al., 2014).



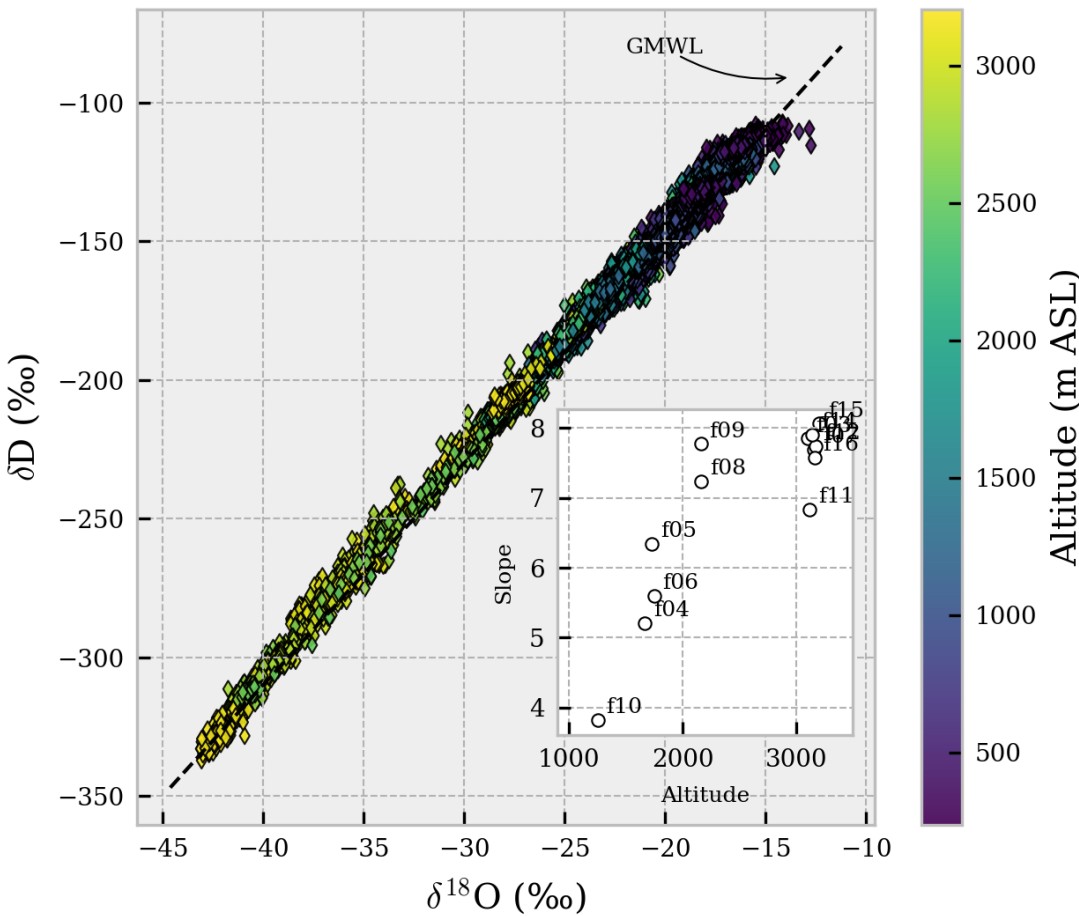

**Figure 8: Distribution of the observations for all the flights on the $\delta^{18}O$ vs $\delta D$ space. The GMWL ($\delta D=8*\delta^{18}O+10‰$) is reported for reference. Inset plot: slope of the $\delta^{18}O$ vs $\delta D$ (‰/‰) linear correlation for individual flights as a function of the maximum altitude (m) reached by each flight (r=0.84).**


**Table 2: slopes of the $\delta^{18}O$ vs $\delta D$ linear fit for individual flights (‰/‰). Flight extent below blh reported as the percentage of data points collected below the blh for each flight. *Denotes flights which flew over an area > 20 km². Correlations reported between [brackets].**

| Flight (ID) | Flight extent below blh (%) | Slope for full flight | Slope for subset < blh | Slope diff. \|blh - full\| |
|---|---|---|---|---|
| f03* | 39.47 | 7.85 [0.98] | 4.74 [0.82] | 3.11 |
| f04 | 85.59 | 5.21 [0.89] | 4.84 [0.87] | 0.37 |



| | | | | |
|---|---|---|---|---|
| f05 | 68.02 | 6.34 [0.91] | 6.63 [0.92] | 0.29 |
| f06 | 63.33 | 5.60 [0.83] | 3.10 [0.66] | 2.50 |
| f07* | 37 | 7.69 [0.99] | 1.75 [0.44] | 5.94 |
| f08* | 64.37 | 7.23 [0.98] | 5.6 [0.94] | 1.64 |
| f09* | 57.22 | 7.77 [0.98] | 3.78 [0.81] | 3.99 |
| f10* | 89.3 | 3.82 [0.75] | 3.08 [0.68] | 0.74 |
| f11* | 53.91 | 6.83 [0.94] | 5.39 [0.86] | 1.44 |
| f12 | 40.59 | 7.74 [0.99] | 4.31 [0.75] | 3.43 |
| f14* | 34.88 | 7.90 [0.99] | 6.89 [0.97] | 1.01 |
| f15* | 29.63 | 8.06 [0.99] | 7.02 [0.96] | 1.04 |
| f16 | 31.83 | 7.57 [0.99] | 7.56 [0.95] | 0.01 |

### 380  3.4 The vertical and horizontal variability of the isotopic composition of water vapor

Two types of flight patterns were used to investigate the vertical and spatial (horizontal) variability of the 3D water vapor isotopic signal in detail. Flights 4-7,11,12, 16 were selected to probe the vertical variability while flights 8-10,14, 15 were selected to probe the horizontal variability. Specifically, flight 9 was designed to investigate the spatial variability during the same flight at two different altitude levels. Flights 8-10 were performed over the Aubenas area (a region of small hills and

mountains), and flights 14 and 15 were performed over the Rhône Valley, near the town of Montélimar. In this section, only $\delta D$ is reported in Fig. 9, because remote sensing technologies, like LIDAR and satellite instruments target the $H_2^{16}O$ and $HD^{16}O$ absorption bands but not $H_2^{18}O$ ($\delta^{18}O$ and d-excess maps are provided in Supplementary Material SM4). The $\delta^{18}O$,





δD, and d-excess variability are now first characterized by the span (max-min) and the standard deviation of the observed δ¹⁸O, δD, and d-excess distributions (Table 3). The isotopic variability is larger for vertical profiles than for horizontal scans

performed at the same altitude, as one might expect from the vertical temperature and humidity gradients. The ratio between vertical and horizontal span is 2:1 for δ¹⁸O, δD, and d-excess, while the ratio between standard deviations is 3:1, 4:1, and 1:1 for δ¹⁸O, δD, and d-excess, respectively. In combination, these ratios highlight δD as the most sensitive parameter in both directions. For vertical flights, the correlation between standard deviation and vertical flight extent is high for δ¹⁸O, δD, and d-excess (0.65, 0.66, and 0.40, respectively). It is worth noting that low-altitude vertical profiles, mostly limited within the

boundary layer, show similar isotopic variability in terms of span and standard deviation. Similar correlation can be observed for δ¹⁸O, δD and flown-over area for horizontal pattern flights but no significant correlation was observed for d-excess. Similar to other studies, this dataset also shows a good correlation between the water vapor isotopic composition (δ¹⁸O and δD) and the logarithm of the specific humidity. Hence, the δ values were modelled using a linear regression model with log(q) as the sole predictor allowing to explain more than 90% of the δD variability during vertical flights. Notably, for flight

7, a high-altitude sounding of the atmosphere, log(q) can explain over 99% of the δD variability. For horizontal flights, the explained variance is smaller but still high on average ($r^2_{\delta D \text{ vs } q}$ = 0.74). Tables reporting all the r² values are provided in the Supplementary Material SM5. Even though the δ-log(q) COSMO$_{iso}$ vertical patterns are consistent to the observed vertical patterns, there is a clear difference between the best-fit parameters for horizontal and vertical flights in the model. Indeed, the average slopes of the δD vs log(q) model estimated from observations are 69.4 and 68.9 for vertical and horizontal

flights, respectively, while the average slopes estimated from COSMOiso output are 65.8 and 122.2.





**Table 3: Span (max-min) and standard deviation of for flights selected to probe the vertical and the horizontal variability of the water vapor isotopic signal. *Denotes vertical profiles with number of observations within blh >50%. All values in ‰.**

| Flight | | | |
|---|---|---|---|
| **Vertical pattern** | **$\delta^{18}$O span (SD)** | **$\delta$D span (SD)** | **d-excess span (SD)** |
| f04* | 3.6 (0.6) | 18.4 (3.4) | 21.1 (2.2) |
| f05* | 6.9 (0.8) | 26.0 (5.4) | 40.7 (2.6) |
| f06* | 4.6 (0.6) | 20.1 (3.7) | 24.2 (2.5) |
| f07 | 23.6 (7.0) | 173.4 (54.4) | 24.6 (3.0) |
| f11* | 5.9 (1.1) | 33.6 (8.1) | 29.9 (3.2) |
| f12 | 11.6 (3.1) | 80.9 (23.8) | 22.5 (2.2) |
| f16 | 12.2 (2.7) | 83.3 (20.7) | 23.5 (2.5) |
| **Average** | 9.8 (2.3) | 62.2 (17.1) | 26.6 (2.6) |
| **Horizontal pattern** | | | |
| f08 | 4.1 (0.7) | 22.8 (4.3) | 15.8 (2.2) |
| f09 | 2.4 (0.5) | 11.0 (2.0) | 13.7 (2.1) |
| f10 | 3.2 (0.4) | 17.8 (1.8) | 20.2 (2.3) |
| f14 | 6.7 (1.1) | 47.3 (7.8) | 18.4 (2.4) |
| f15 | 6.9 (0.9) | 51.9 (6.8) | 14.9 (2.1) |
| **Average** | 4.7 (0.7) | 30.2 (4.5) | 16.6(2.2) |

## 3.5 The vertical and horizontal spatial structure of the isotopic composition of water vapor

Since the water vapor isotopic composition is strongly correlated with the specific humidity (and consequently with air temperature), the variogram of the residuals of the linear model defined between log(q) and δ values enabled the investigation of the spatial correlation of different isotopologues of water vapor alone. The variograms for $\delta^{18}$O, $\delta$D and d-



excess for both flight patterns are shown in Fig. 9. A spherical model was used to fit the observed semivariance within a maximum lag-distance of 5 km. The same procedure was applied to COSMO$_{iso}$ output. Even though each flight presents a specific pattern, some general observations can be made. First, a large part of the variance in isotopes can be explained by the variability of the specific humidity and the average variability of model residuals is only ~0.5‰, ~2.8‰, and ~2.3‰ for $\delta^{18}$O, $\delta$D, and d-excess, respectively (the sill values for observations in Fig. 9). Such values are only slightly larger than instrumental precision and must therefore be interpreted carefully. In this context, it is clearly visible that the average variograms computed on observations and those from COSMO$_{iso}$ output are offset by ~0.3‰, ~1‰, and ~2‰ at 0 m distance (i.e., the nugget values), consistent with the values attributed to instrumental uncertainty (0.23‰, 0.50‰, and 1.78‰ for $\delta^{18}$O, $\delta$D, and d-excess, respectively). Secondly, the spatial structure extrapolated from observations differs between vertical and horizontal flights. This spatial anisotropy is especially noticeable for $\delta$D, as highlighted in section 3.4, and the COSMO$_{iso}$ model seems to not capture such anisotropy. Finally, the spatial correlation of the model residuals acts over a short range, averaging ~1000 m for both $\delta^{18}$O and $\delta$D in observations. The range for d-excess is limited to less than 250 m in observations and ~1300 m in COSMO$_{iso}$. Given the limited variability in d-excess, however, it is not possible to formulate more detailed hypotheses about this parameter.





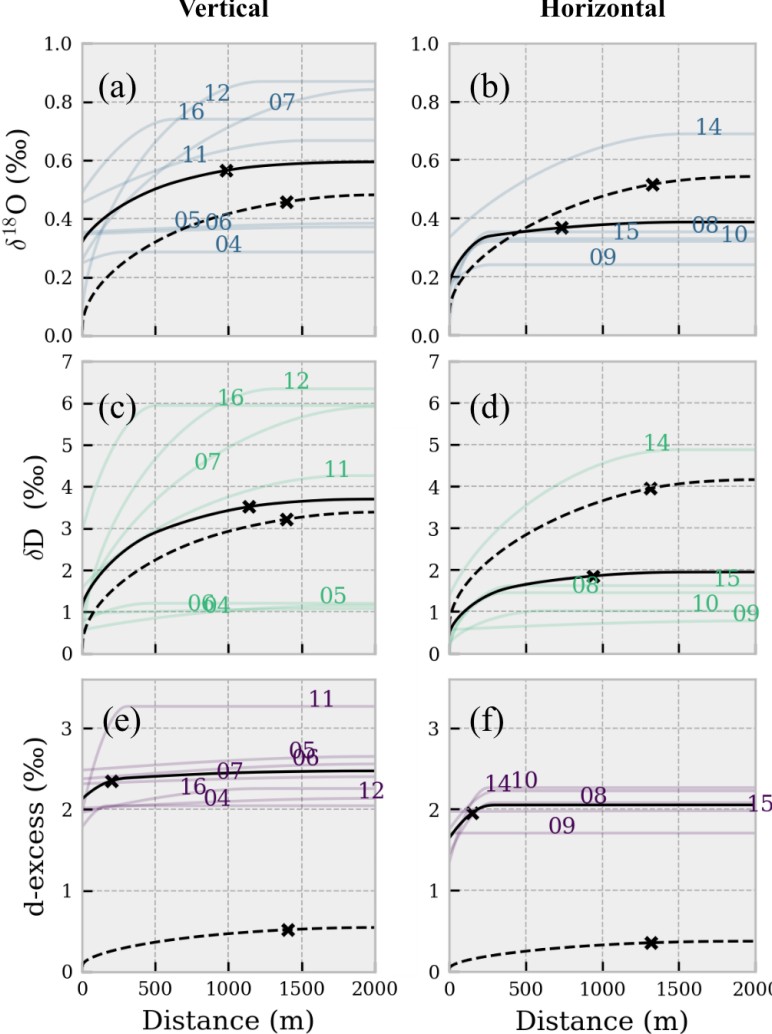

**Figure 9: Square root of the semivariance of the $\delta$ vs log($H_2O$) model residuals as a function of the distance. $\delta^{18}O$ (a) and (b), $\delta D$ (c) and (d), d-excess (d) and (e). The colored lines represent the square root of the spherical model variograms estimated for each flight. Solid black lines are the ensemble means considering all the flights of the panel. Dashed black lines are the ensemble means calculated on COSMO$_{iso}$ output interpolated on flight paths (variograms for each flight are not reported to improve visual interpretation). The "x" on the ensemble mean curves denotes the average distance at which residuals are uncorrelated (95% of the sill).**

Focusing on the observations, the vertical variograms in Fig. 9 show a striking difference between low altitude and high-altitude flights (flights 4,5,6 and flights 7,11,12,16). Hence, the spatial correlations for vertically resolved observations of water vapor isotopic composition is stronger the larger the atmospheric column probed is. This is reasonable, since different height levels can be representative of different large-scale circulation and therefore can be imprinted by water vapor with different isotopic signatures. Flight 10 provides insights on how the spatial pattern of water vapor isotopic composition is



sensitive to the fine-scale (<100 m) process, as further discussed in section 3.6. For horizontal flights on single level, all the flights but fight 14 show a similar pattern in spatial structure. As can be noted from Fig. 8, flight 15 is almost a replica of flight 14 in terms of flight pattern, location and altitude level. However, flight 14 was performed in the morning and flight 15

in the early afternoon. The key differences between these two flights are further discussed in section 3.7.

## 3.6 Water vapor isotopes spatial patterns at different altitudes

Now we analyse the fine-scale horizontal structures in the variations of the stable isotope composition across different levels of the boundary layer targeted during specific flights. The second part of flight 10 consisted in the spatial sampling of the atmosphere at three different altitudes in the boundary layer near the Aubenas Aerodrome: $763 \pm 12$ m, $917 \pm 13$ m, $1229 \pm 8$

m, hereafter L700, L900, L1200 (Fig. 10.a). Each level was probed for 20-30 minutes and covered a horizontal scale of 6.1 x 2.8 km. A well-mixed atmosphere and low variability of $\delta$D can be observed within the boundary layer, as shown in Fig. 10.c and Fig. 10.d. The small-scale variability of $\delta$D and q is reflected by the low $r^2$ for the $\delta$D vs log(q) regression model of horizontal scans at L700 and L900 (0.53 and 0.55, respectively).

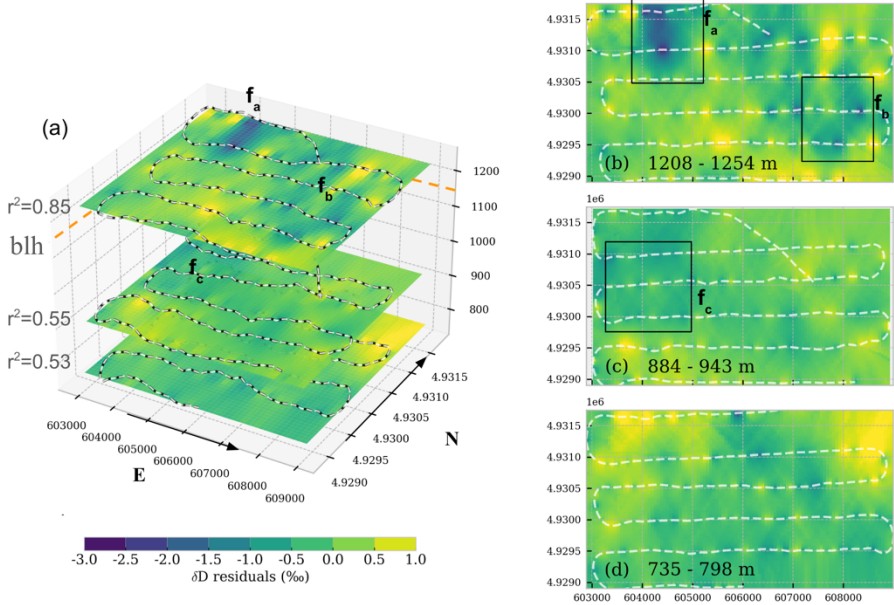

**Figure 10: Residuals field of the $\delta$D vs log(q) model at different altitudes during flight 10 obtained by ordinary kriging. (a) Stacked view of levels L1200, L900 and L700 at average altitude level (1229, 917, 763 m ASL). The orange dashed line indicates the boundary layer altitude (1120 m ASL). (b-d) Details of residuals fields for each level. The text reports the min-max altitude recorded by ULA for that level. For all panels, the zebra-style lines indicate the ULA path. Areas marked with $f_x$ are discussed in the text. Horizontal projected coordinates are in meters (WGS84 UTM zone 31).**


At L1200, close to the boundary layer height, the $r^2$ significantly increases (0.83) and spatial features in the residual field are more evident (Fig. 10b). The non-random spatial structure of residuals is confirmed by Moran I, which is statistically





significant for all the three altitude levels, and it is the highest for the top level (I=0.44, p-value < 0.01, estimated with a distance band of 250m). More specifically, the features $f_a$ and $f_b$ highlight short living and size limited processes that are

characterized by more depleted water vapor than predicted by the $\delta$D vs log(q) relationship. These coherent features are not related to water vapor analyzer performances, since no correlation was observed between model residuals and instrument performance indicators (e.g. sudden changes in cavity temperature, cavity pressure etc) proving that such features are measurable changes in the water vapor isotopic composition. Another proof of the presence of such spatial features is given by the fact that each feature is probed by the ULA at least two times, with opposite cruise direction. Interestingly, there is no

apparent direct link between spatial features at the different levels observed. For instance, feature $f_c$ on L900 cannot be easily associated to feature $f_a$ on L1200, meaning that such features are highly resolved on the vertical axis and spreaded over the horizontal plane in the order of ~1 km. Therefore, we speculate that the ULA may have captured intermittent coherent structures which are commonly observed at the boundary layer top over terrain with high surface roughness (Thomas and Foken, 2007).


### 3.7 Temporal evolution of water vapor isotopes spatial patterns

Flights 14 and 15 were designed to probe the spatial variability of water vapor isotopic composition above the Rhône Valley at different times during the day, as shown in. Fig.11. Notably, both flights 14 and 15 are characterized by large spatial autocorrelation (Moran I = 0.87 and 0.72) but flight 14 is characterized by the strongest spatial autocorrelation structure

among all the horizontal pattern flights (see Fig. 9).

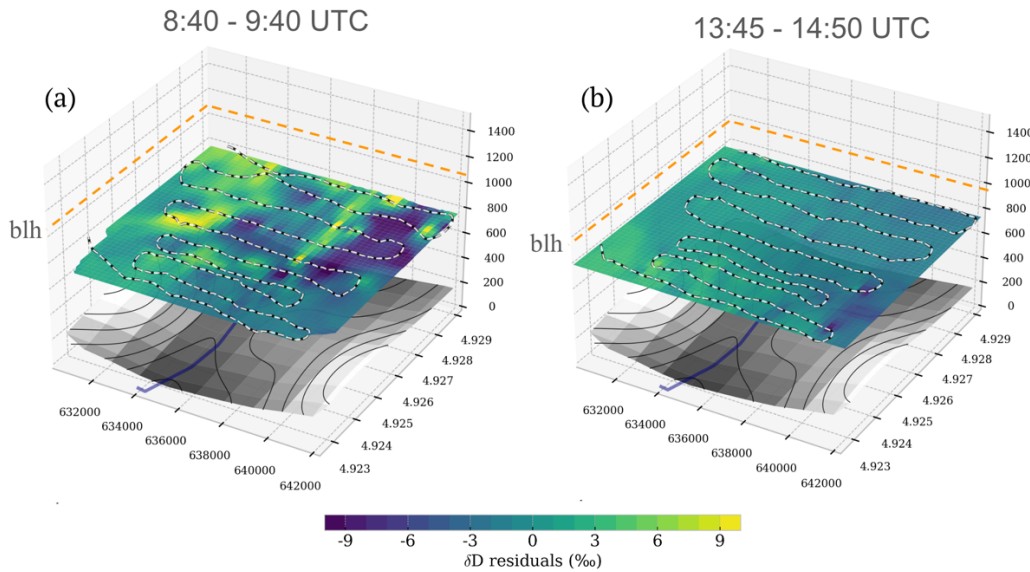

**Figure 11: Residuals field of the $\delta$D vs log(q) model obtained by ordinary kriging for the same location above the Rhône Valley at different times of the day: (a) morning flight 14, (b) afternoon flight 15. Colors, units and lines format like Fig. 10. Underlying topographyand the Rhône River are reported for reference. Horizontal projected coordinates are in meters (WGS84 UTM zone 31). Vertical axes are in m ASL.**






A few hours later, flight 15 shows that the same area is characterized by a less evident spatial structure, which is similar to the one observed for all the other horizontal pattern flights. As briefly shown on the three layers of flight 10, the more evident the spatial features in the residual fields are, the smaller the $r^2$ of $\delta D$ vs $\log(q)$ is ($r^2 = 0.53$ and $0.90$ for flight 14 and flight 15, respectively). Following the underlying topography, it is possible to see that the simple specific humidity estimate reveals larger positive deviations on the west side of the map, where the morning sun very likely produced unevenly heating of the Rhône Valley, promoting the formation of a thermal on the east-exposed slopes and accentuating the signal of surface evaporation the isotopic composition of water vapor (being the evaporation flux enriched with respect to ambient moisture).

### 3.8 Simulating the vertical variability of water vapor isotopic composition

Having seen that water vapor mixing ratio can provide a first-order approximation of the vertical and horizontal water vapor isotopic structure in the atmosphere, we will see here how conceptual models, based on humidity only, would deviate from expectation in terms of water vapor isotopic composition. As described for the observational data in section 3.2, the specific humidity, water vapor isotopic composition, and air temperature were binned and averaged over 20 height levels with 150 m vertical resolution for each flight. The squared difference (error) between modelled $\delta^{18}O$, $\delta D$, and d-excess and the bin-averaged observations was used as a metric to evaluate the performance of the conceptual models.

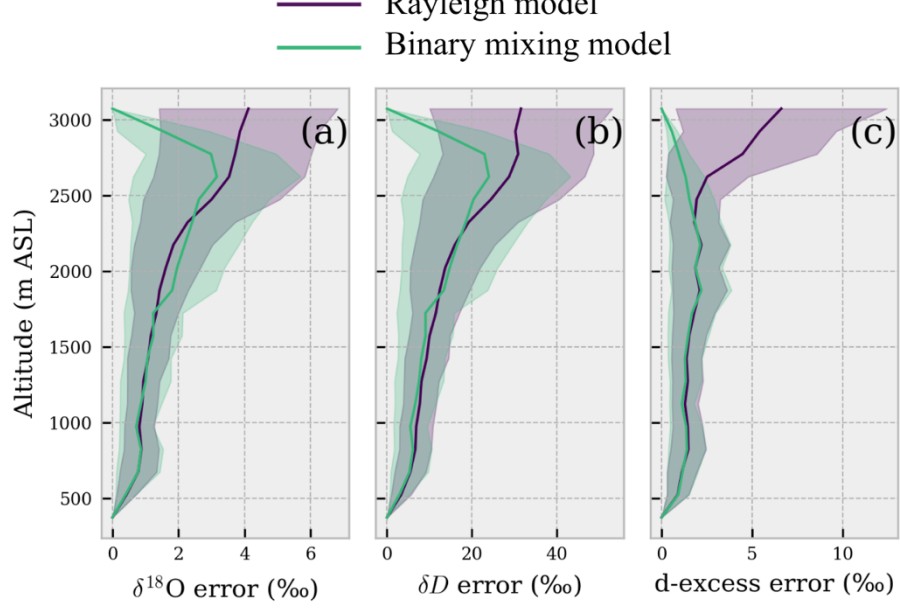

**Figure 12: Root Mean Square Error (RMSE) between models and observations averaged per height levels for $\delta^{18}O$ (a), $\delta D$ (b) and d-excess (c). The solid lines represent the average error calculated over a 150m bin size for all the flights and shadings represent the standard error of the mean.**

In general, both models can predict the variability of water vapor isotopic composition to a reasonable degree, as shown in Fig. 12. The actual modelled vertical profiles compared to observations are available in the Supplementary Material SM6.





Globally, considering all flights and vertical levels, the Root Mean Squared Error (RMSE) varies within narrow ranges: [1.5 - 1.8] ‰ for δ¹⁸O, [11 - 15] ‰ for δD, and [1 - 2] ‰ for d-excess. Both conceptual models achieved very similar results within the boundary layer (<1000 m ASL). However, it is worth noting that even though both models produce similar results, the Rayleigh model is in principle less suited to explain the processes of a strongly mixed and turbulent boundary layer, where there is water vapor mixing between the free troposphere and surface evaporation flux, as suggested e.g. in Benetti et al. (2018) for marine environment. This hypothesis is partially supported by the fact that the binary mixing model generally performed better than the Rayleigh model. Indeed, the Rayleigh model should be better suited to describe the development of a convective cloud, which was not the case for most of the flights in this study except for flight 11, which was specifically designed for sampling water vapor above and below (but not within) a convective cloud. Nevertheless, results show that water vapor isotopic observations measured above 2500 m are challenging to capture for both the Rayleigh and mixing models, as both methods yield large errors for δ¹⁸O and δD. Similar results are obtained using COSMO$_{iso}$ as reported in Supplementary Material SM7. The mixing model performs better than the Rayleigh model in simulating d-excess, although the differences between the two models are small. The mixing model shows a smaller RMSE (~1‰) and a d-excess error distribution that is consistent across different height levels. Further, the error for the Rayleigh model is more spread out above 2000 m ASL. The analysis of d-excess profiles for individual flights reveals that the shape of Rayleigh-simulated profiles is almost flat below 2500 m ASL (not shown), which is expected because d-excess variability is small during equilibrium fractionation in the Rayleigh distillation process. The d-excess simulated with the mixing model follows the general trend of observed d-excess within the vertical profile.

## 4 Discussion

### 4.1 Spatial representativeness of water vapor isotopic composition in the atmosphere

As shown here and in several other studies, the log of specific humidity and the water vapor isotopic composition are strongly correlated (e.g. Lee at al., 2007; Sodemann et al., 2017). Therefore, the spatial representativity of water vapor isotope observations is intrinsically related to spatial representativeness of water vapor mixing ratio to a first order (if dominated by turbulent mixing). The spatial correlation scale of the atmospheric water vapor is a quantity that depends on the turbulence conditions of the atmosphere and on the weather regime among other factors. Therefore, the spatial representativeness of specific humidity can exhibit patterns across different spatial and temporal scales. In this study we observed that the semivariance of specific humidity at a given spatial separation estimated from horizontal pattern flights at different altitudes tends to continuously increase as function of the distance, and no observable plateau can be identified within a radius of 5000 m (see Supplementary Material SM8). Hence, 2 and 10 km resolution COSMO$_{iso}$ lowest level data was used to replicate a similar analysis on a large area (3˚x4˚) centered over Aubenas. The results in Fig. 13.a, extrapolated at the same time of the flights, reveal the occurrence of one or more plateaus for specific humidity at different separation distances, depending on the model resolution. As a further control, the same analysis was performed on the specific humidity



of ERA5 at the lowest pressure level, confirming that a first plateau can be identified between 100 - 300 km, varying from
day to day (data not shown). The results reported in this study agree with the findings by Park et al. (2018) which report drop
in spatial correlation for water vapor concentration at a separation distance > 100 km. As expected, similar results in term of
separation distance and drop in spatial correlation are obtained for $\delta$-values and d-excess (Fig. 13.b and c, the observed
semivariance pattern in this study is similar for $\delta^{18}$O and $\delta$D and is not reported here). Similar separation distance (300 km)
has been also used by Thurnherr et al. (2024) to obtain total column averaged $\delta$D retrievals from S5P satellite in southern
France. In conclusion, 100 km can be considered an approximate threshold for collecting statistically independent water
vapor isotope observations when considering processes acting on the mesoscale.

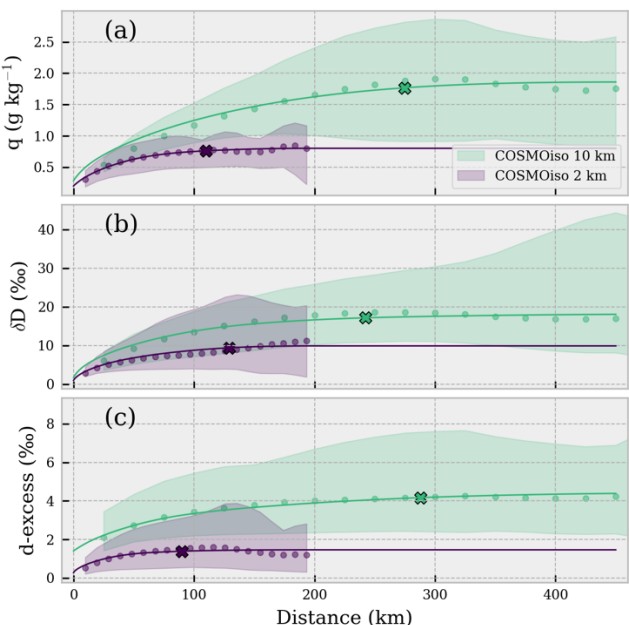

**Figure 13: Similar to Fig. 9, the square root of the semivariance of q, $\delta$D and d-excess (a, b and c, respectively). For all panels:**
**colors are representative of model runs at different resolutions, dots are average experimental variogram, solid lines and shadings**
**represent ensemble mean and min-max interval of the square root of the spherical model variogram. The "x" on the ensemble**
**mean curves denotes the average distance at which residuals are uncorrelated (95% of the sill).**

**4.2 Stable isotopes of water vapor highlight fine scale processes**

When the covariance between the humidity and its isotopic composition is accounted through simple linear regression, or by
means of conceptual models, fine scale processes can be detected by fast and localized changes of the isotopic composition
of water vapor alone. The example of flight 10 shown in section 3.5 highlights how quickly the autocorrelation of the water
vapor isotopic composition drops as a function of distance. Such autocorrelation can also change quickly as a function of
time depending on changes in wind speed and thermodynamic conditions within the boundary layer. For instance, flight 14
and 15 in section 3.6 showed that differential heating due to topography, likely introducing the development of thermals, can





produce significant changes in the water vapor stable isotopes field. Our results hence suggest that water vapor isotopes could be used as a proxy for studying boundary-layer development, including turbulent mixing processes and the role of coherent structures for the exchange between the boundary layer and the free troposphere. However, technical issues might arise studying such water vapor isotopic composition at such a small temporal scale due to the slow response time and the memory effect in CRDS current measurement technology. Thus, optimal filtering of isotopic signals as proposed in section

2.4 is paramount when using a fixed 2-levels keeling plot with roughly hourly time scale to determine accurately the isotopic composition of the ocean evaporation flux (Steen-Larsen et al., 2014, Zannoni et al., 2022) and evapotranspiration (Aemisegger et al., 2014). Further corrections are indeed necessary when fluxes are estimated at even higher frequency, such as with eddy covariance - CRDS coupled systems (Wahl et al., 2021). The recent work by Meyer and Welp (2023) highlights that flow rate and optical cavity volume are indeed key factors contributing to the overall memory effect in laser analyzer. In

addition to this, we suggest using a short inlet, low-memory inlet material (e.g., polished or coated stainless steel, copper), suitable heating or insulation, and fast flow rates when performing high-frequency measurements. We also emphasize the need for a dedicated study to identify the best materials and optimized high flow rate settings for water vapor isotope flux analysis, which would greatly benefit the isotope-hydrology community.

**4.3 Vertical representativity of water vapor isotopic composition in the atmosphere: extrapolation of $\delta$D for the full column**

The results of this study depict a limited variability in water vapor isotopic composition in the horizontal space and a large variability in the vertical direction. Such a variability accounts roughly for a 1:4 ratio, based on $\delta$D standard deviations, which might be sensitive to measurement uncertainty and to the shape of the isotope data distributions. As mentioned before,

the large vertical variability is not surprising given the large temperature and humidity gradients in the atmospheric column. However, the results of the comparison between the conceptual models and ULA observations suggest that a few data points within the boundary layer can be used to estimate the vertical profile of the water vapor isotopic composition up to several km with a certain degree of confidence. Despite the results in section 3.4 indicating vertical turbulent mixing as the main controlling process of the water vapor isotopic composition in the lower troposphere, the quantities involved in such

idealized two-endmembers model are not straightforward to predict. Most important, information about the average water vapor isotopic composition of the free atmosphere ($\delta_0$) and about the isotopic composition of the surface flux ($\delta_F$) are required terms in the mixing equation. For example, we estimated a change from $\delta^{18}O_F$ = -5.25‰ at 5 UTC to $\delta^{18}O_F$ =- 13.11‰ at 15 UTC on 18 Sep (flights 4 to 7). The early morning $\delta^{18}O_F$ closely align with the average isotopic composition of precipitation for the study area in September, which is -5.3±2.0‰ using 1997-2022 GNIP data ~100 km south the study

area and accounting for an altitude effect (Masiol et al., 2021). This suggests a significant impact of transpiration on the surface flux during the early part of the day, assuming the composition of the transpiration flux equals to the average composition of precipitation in the study area. The subsequent decrease in $\delta^{18}O_F$ along the day points to an increasing



evaporation contribution to the surface flux. This change in the composition of the flux end member shows that assigning a constant isotopic signature of the surface evapotranspiration flux based on precipitation around the study area is not feasible. 595 The same applies for the variability of the dry end member $\delta_0$, whose composition can only be guessed or measured with dedicated high-altitude flights. It should be noted, however, that the results showed the $\delta$D vs log(q) relationship holding even if the controlling physical process modulating the isotopic composition in the lower troposphere is mixing, which in principle should be represented by an hyperbole in the q-$\delta$ space (the reader is referred to Supplementary Material SM9 for a comparison among observations, Rayleigh distillation and mixing model). Mathematically this can be explained by the fact 600 that a hyperbolic curve can be fitted by a logarithmic curve within a limited range of values.

Focusing on $\delta$D, which can be also retrieved with remote sensing through the atmosphere, the best-fit parameters of the log-linear model $\delta D = \beta_0 * \log(q) + \beta_1$ [‰] for all the flights of this study are $\beta_0 = 93.86$ and $\beta_1 = -324.0$ (see Supplementary Material SM9 for individual best fit parameters of each flight). It is worth noting that the shape of the $\delta$D vs q relationship is similar across different airborne datasets, as shown in Fig. 14 (Chazette et al., 2021, Dyroff et al., 2015, Dryoff et al., 2021, 605 Salmon et al., 2019, Schneider et al., 2015, Schneider et al., 2018, Sodemann et al., 2017, Wei et al., 2019). Supplementary Material 10 shows the resulting plot on a semi-log space.

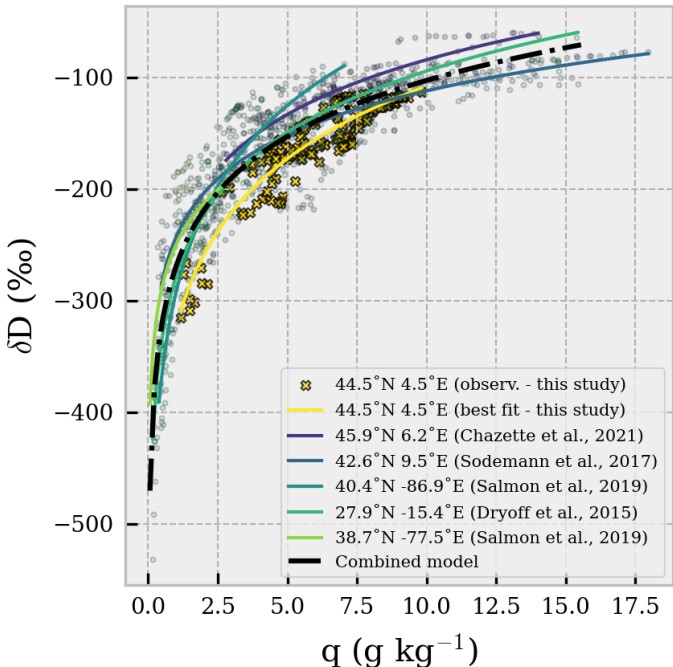

**Figure 14: $\delta$D vs q over 150 m binned vertical profiles estimated for different airborne campaigns. The legend reports the coordinates of the flights and the reference study. Symbols are observations, solid lines are best-fit curves. The black dot-dashed** 610 **line is the best-fit curve combining all the binned vertical profiles from all the datasets. The best fit model for all the curves is $\delta D = \beta_0 * \log(q) + \beta_1$.**



Indeed, $\beta_0$ shows small variability, ranging from 70.62 (Annecy, Chazette et al., 2021) to 103.96 (Indianapolis, Salmon et al.,
2019). When all the observations are combined $\beta_0 = 72.31 \pm 0.94$, where the uncertainty is the standard error of the slope.
Similarly, the $\beta_1$ parameter, ranges from -324.0 to -243.1 (yielding $\beta_1 = 269.4 \pm 1.6$ for all combined observations). Such a
limited variability in the best-fit parameters highlights that the log-linear approximation of the mixing process holds its shape
across different locations and for different vertical extents of the tropospheric column probed with each flight. Changes in
the weather conditions, such as, strong/weak convection, strong/weak entrainment, atmospheric stratification, presence of
clouds, etc. are likely to affect the *shape* parameter ($\beta_0$). Changes in the isotopic composition of the two endmembers of the
binary mixing (i.e. the water vapor in the boundary layer and in the free troposphere) are likely to affect the intercept
parameter ($\beta_1$).

The main advantage of such a log-linear approximation is that just a single level observation of $\delta$D and the tropospheric
humidity profile are necessary to produce an approximation of the tropospheric profile of water vapor $\delta$D in clear sky
conditions. This in turn can be used to estimate the weighted average water vapor column $\delta$D, providing information on the
total column water vapor $\delta$D (assuming the measured humidity profile captures ~100% of the total column water vapor).
Following this approach, the single level observation can be surface observations of water vapor isotopic composition that
are representative for the boundary layer. The vertical distribution of the water vapor mixing ratio can be retrieved with
regular vertical profiling such as radiosounding. To scale the log-linear model for a specific location and time, the model can
be rearranged in the form:


$$\delta D = \beta_0 \log\left(\frac{q}{q_{SURF}}\right) + \delta D_{SURF} \quad (2)$$

where $\beta_0$ is the best-fit parameter reported above (72.31 ± 0.94), $q$ is specific humidity profile [g kg$^{-1}$], $q_{SURF}$ is the mixing
ratio measured at the surface [g kg$^{-1}$] and $\delta$D$_{SURF}$ is the water vapor $\delta$D measured at the surface. Figure 15 shows the
distribution of the differences between modelled and observed weighted average water vapor column $\delta$D considering all the
datasets used to generate Fig.14. The mean difference between observed and modelled weighted average $\delta$D is 4.2 ± 12.7 ‰
(n = 59). However, when considering only flights which probed the troposphere for a vertical extent of at least 5000 m ASL,
the difference becomes 12.2 ± 6.7 ‰ (n = 6, all flights from Dyroff et al., 2015). On average, the log-linear model returns
negatively biased $\delta$D values. The Root Mean Squared Error between observed and modelled weighted average $\delta$D can be
representative of the uncertainty of the log-linear model approximation, being also very similar when using all the datasets
and when using only datasets with flights >5000 m ASL (13‰ and 14‰, respectively). It is worth noting that with the
simple generalization of the log-linear model important processes such as advection and cloud formation can be easily
missed. Hence, model extrapolations should be approached with caution, and a clearer understanding of the factors
influencing the β0 and β1 parameters is essential to provide an initial approximation of the $\delta$D profile for potential satellite
validation. Simultaneously, this exploratory analysis highlights the value of incorporating stable isotopic composition of

 

water vapor to improve parameterization of atmospheric hydrological processes, which may be less accurately captured by variations in specific humidity alone, as demonstrated by numerical weather forecast simulation experiments (Yoshimura et al., 2015; Toride et al., 2021).

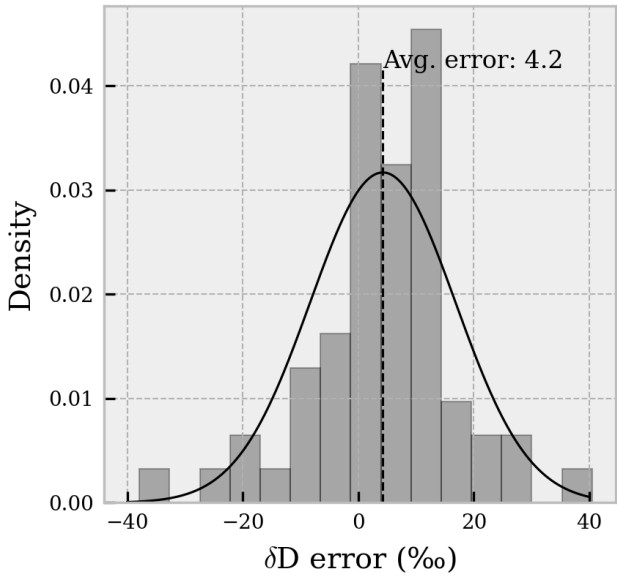

**Figure 15: Error distribution (Observed - Modeled) of the estimated weighted average atmospheric $\delta$D. Solid black line represents a normal distribution with mean = 4.2‰ and standard deviation = 12.7‰**

**5 Conclusions**

In this study, we used a highly temporal and spatially resolved airborne dataset in combination with conceptual and numerical models (COSMO$_{iso}$) to gain insights into the controlling factors of water vapor isotopic composition in the lower troposphere and its spatio-temporal representativeness. Our findings indicate that vertical mixing is the dominant process affecting isotopic variability in the lower troposphere at hourly and sub-daily scales for this study. Within such a temporal scale, significant isotopic fractionation effects, as well as possible advection, become important at altitudes above 3000 meters. At these higher altitudes, both conceptual and numerical models struggle to accurately simulate water vapor isotopic composition. Interestingly, our flights combined data perfectly align with the Global Meteoric Water Line (GMWL), unlike typical surface-only studies which often report $\delta$D vs. $\delta^{18}$O slopes smaller than 8. However, the $\delta$D vs. $\delta^{18}$O slope varied by flight, showing a strong positive correlation between the maximum altitude reached by each flight and the slope. Small slope values (< 8 ‰/‰) have been observed mostly within the boundary layer, indicating the influence of local evapotranspiration flux in the lower tropospheric moisture. The increase in slope at higher altitudes is due to the larger number of data points at the more depleted end of the mixing curve during higher-altitude flights. The analysis of isotopic composition variability revealed substantial differences in the spatial structure of water vapor isotopes between vertical and horizontal flights,




indicating a clear spati 6al anisotropy for δD. This anisotropy at a distance up to 5000m is not captured by the COSMO$_{iso}$ model. More broadly, the analysis highlighted a large-scale horizontal control of the water vapor δD and δ¹⁸O signals (100-300 km), which can be approximated by a simple δ-log(q) relationship. Instead, the rapid and localized changes in δD and δ¹⁸O 3D fields (1000-1500 m range) underscore the utility of isotopic measurements in studying atmospheric dynamics at the

microscale. Finally, our results provide a first-order approximation of vertical δD variability as a function of the specific humidity, thereby enabling better scaling of surface δD observations to the tropospheric column for improved δD satellite validation. We believe that the dataset and findings of this study will aid future research aiming to combine observations, numerical simulations, and satellite retrievals of water vapor isotopic composition.

**Code/Data availability**

The geolocated observations of humidity, water vapor isotopic composition, temperature and atmospheric pressure acquired with the Ultralight Aircraft are available here https://doi.org/10.5281/zenodo.7864006. The Python code to analyse the data and produce the figures will be uploaded on GitHub-Zenodo.

**Competing Interests**

The authors have no conflicts of interest to declare.

**Acknowledgements**

This research has received funding from the European Union's Horizon 2020 research and innovation program under grant

agreement N° 821868. FARLAB at University of Bergen, Norway, is gratefully acknowledged for supporting this research with their CRDS analyzer and other instrumentation. The authors are grateful to Air Création and Tignes Air Experience for their professional support with the airborne and ground activities during the field campaign.

**Authors contributions**

DZ, HCSL and HS conceptualized the study. DZ together with HCSL, HS, PC, JT and MR carried out the field activities. DZ developed the methodology and investigation. DZ led the formal data analysis, the visualization and data curation. HS carried out the weather situation analysis. IT carried out the COSMO$_{iso}$ simulations. MW provided the ECHAM6-wiso boundary data for COSMO$_{iso}$. MR, CF, HCSL and HS acquired funding for this study and administrated the project. DZ, HCSL, HS, IT, CF, PC, JT, MW and MR contributed to writing the original draft.




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
