# Peer review of "Vertical and horizontal variability and representativeness of the water vapor isotope composition in the lower troposphere: insight from Ultralight Aircraft flights in southern France during summer 2021"

_EGUsphere, 2024_

## Author Comment (AC1)

Reply to

RC2: 'Comment on egusphere-2024-3394', Adriana Bailey, 08 Mar 2025

In the following text the Referee's (#1) comments are reported as normal lead text and authors answers are reported in grey shaded box like this one with *italic font*. The edited text in the manuscript is in *red color*.

**Vertical and horizontal variability and representativeness of the water vapor isotope composition in the lower troposphere: insight from Ultralight Aircraft flights in southern France during summer 2021**

This study uses unique ultralight aircraft measurements in southern France to examine the horizontal and vertical variability of the stable water vapor isotope composition in the boundary layer and in the lowermost free troposphere. Evidence of coherent isotopic structures on the scale of 100s of meters are identified and discussed. The study also finds that vertical mixing is the dominant process that influences the boundary-layer water vapor isotope ratio on short timescales. A simple two-parameter model for estimating the isotopic profile of the boundary layer is presented based on these results.

*We are grateful to Referee #1 for her valuable comments and insightful suggestions, which have helped improve our manuscript. We believe we have addressed all the Referee's comments. Below, please find our detailed, step-by-step responses*

The technical elements of this work are fascinating. And, the analysis is thorough and well structured, but the paper is also dense and long. I feel this work would reach a larger audience if the importance of identifying isotopic coherent structures could be conveyed more clearly. Indeed, a "key point" for each subsection would go a long way in helping bring the reader along with the manuscript.

*We have reduced the length of the manuscript where possible, particularly in the methods section at the following locations (line numbers refer to the original preprint):*

*Lines 122-126: moved to Supplementary Material SM0. Main text in the manuscript edited as follows:*
*The reader is referred to Supplementary Material SM0 for details on frequency of usage, values of isotope standards and calibration performances of the CRDS analyzer.*

*Lines 197-205: Rephrased in a more synthetic way*

*Regarding the isotopic coherent structures, we have expanded the discussion by estimating their characteristic size and lifetime (with references to previous studies), and their significance (along with their limitations) for studying boundary layer development*

*and turbulent mixing processes (Section 4.2).*

*To address the request for key points in each subsection, we have revised the subsections titles to better reflect their main messages and to guide the reader through the manuscript.*

*For example:*

- *section 4.2 "Stable isotopes of water vapor highlight fine scale processes" becomes "Stable isotopes of water vapor highlight fine scale processes and coherent structures of the water vapor field: current limits using CRDS analyzers"*
- *section 4.3 "Vertical representativity of water vapor isotopic composition in the atmosphere: extrapolation of dD for the full column" becomes "Vertical Representativeness: to what extent do surface observations reflect water vapor isotopic composition in the atmospheric column? Toward a simple extrapolation of $\delta D$"*

So too, I think the paper requires some discussion on the generalizability of the findings to other atmospheric conditions and climatic environments. Figure 14 begins to address this question but only scratches at the surface: the flights considered represent a limited zonal band, and descriptions of the flight conditions (e.g. are they convective? quiescent?) are missing.

*As suggested by the reviewer, we have also expanded the discussion on the generalizability of our findings in Section 4.3, particularly about the vertical representativeness of $\delta D$ in water vapor from low-level observations. Specifically:*

- *We emphasize the limitations of our study, particularly the limited comparison with observations over oceanic and polar regions.*
- *We briefly discuss the impact of different meteorological conditions, referencing flight conditions in both our study and previous works.*

*We believe this revision clarifyies the exploratory nature of this section while maintaining the primary focus of the manuscript on the spatial variability of isotopic signals in water vapor at small scales. We hope this study provides further evidence that water vapor isotopes can serve as valuable tracers for investigating atmospheric turbulence and cloud formation processes, as they reveal small-scales atmospheric structures beyond humidity variations alone.*

Additional, specific suggestions follow.

L 37 - To say that a process affects the stable isotopic ratio of water at the molecular level is a bit odd. The isotope ratio is necessarily a bulk feature of a large collection of molecules, it is not a property of a single molecule.

*We agree the sentence can be easily misunderstood, hence we rephrased as follows:*

*"Stable water isotopes are valuable for studying atmospheric water processes because phase changes influence their isotopic ratios through isotopic fractionation…"*

L 56 - "Specifically, the extent to which water vapor concentration and isotopic composition can resolve different atmospheric processes is still unclear." Does this mean in terms of measurements or expectation? I would argue that theory largely guides us in what to expect and that the question of interest is whether we can measure it.

*As pointed out by the referee, theory can serve as a basis to understand the order of magnitude of processes across temporal (and spatial) scales, as e.g. well described in Dee et al. (2023). Here we refer to the low number of data/studies aimed at resolving short spatial and time scale processes using stable isotopes: e.g. eddy flux measurements, highly resolved atmospheric sounding, etc. We keep the reference to the study of Graf et al. (2019) because we believe is a great example of subcloud raindrop evaporation elucidated with coupled rain-vapor observations.*

*The sentence has been edited as follows:*

*"However, uncertainties remain regarding the control of water vapor isotopic composition in the lower troposphere at meso- and microscales due to the limited number of resolving sub-hourly processes (e.g. Aemisegger et al., 2015, Graf et al., 2019), even though water cycle physics and isotope theory can provide insights on expected patterns."*

L 62 - I don't agree that satellites generally provide breakthroughs on small scale and short lived processes. Their large footprint and infrequent sampling typically make them better suited for larger scale questions.

Yes, this indeed might sounds in contrast with what "small scale" means along the manuscript. The sentence has been edited as follows.

*Remote sensing on satellites can provide important large-scale data that can serve as background for further small-scale investigations, providing nearly global coverage of H2O and HDO pairs at daily resolution*

For Fig1, I suggest adding place labels for the key locations referenced later in text.

*Done. LFHO, Rhône river and area of study cases (f10,f14,f15) are now visible in the figure.*

Section 2.4 and Caption of Figure 3 could be made more accessible and clear. What is the key message of this subsection? I assume the data are corrected for the time response, but this is not actually stated. Also, how does the time response change with water vapor concentration?

*Thank you for highlighting this point, we believe this is an important section of the*

*methods and we want to be sure it is 100% clear.*

*When using high frequency (and then for a moving measurement system, highly spatially resolved) data, a multi-spieces CRDS gas analyser will provide observations with slightly different timing for each isotope. The key message here is that if different instrumental timings are not addressed or the signal is not corrected for, artifacts can emerge, especially in combined signals! E.g. d-excess or q-$\delta D$ in our case.*

*Section 2.4 shows the approach we used for correction, which consists on the design of an Optimal Filter based on the impulse response of the measuring system.*

*Yes, the data presented is corrected for time response and we now clearly states at the end of section 2.4:*

*The data used in this study is corrected as described above, and corresponds to fields with "_OF" extension in Zannoni et al. (2023) dataset, where both uncorrected and corrected measurements are available.*

*About the time response at different humidity concentration: we haven't investigated the impact of humidity concentration on the time response in the field because it would have required time consuming effort that would have interfered with field measurements. However, we believe that the time response we estimated can be also representative of low humidity conditions, even though some effects might still have a small impact, because of the different absorbtion/desorbtion properties in tubing and optical cavity as funcion of the water vapor pressure. We believe this aspect could be further investigated with dedicated laboratory experiments.*
* * *
L340 - The paper states "blue and orange circles" but possibly blue and light green are meant instead?

*The Referee is correct, the text refers to an old version of the plot with different colors. The correct colors for 18 and 22 Sep are blue and light green. The text has been corrected accordingly.*
* * *
L 366- "…Such a positive correlation might indicate the imprint of a local evapotranspiration signal in the boundary layer moisture." It would be helpful to explain why this is the case, since there are two ways in which low slopes can be achieved. Either dD can decrease slowly relative to d18O, causing dD to fall above the MWL, or dD can fall below the MWL as the atmospheric moisture becomes isotopically heavier.

*From isotopic fractionation theory during evaporation, it is likely to say that lower slopes are the result of mixing with surface evaporation flux, where non-equilibrium fractionation processes are important. This is also supported by several other ground based observations, where the slope is almost always (except under saturated conditions) $\leq 7$. Moreover, low level moisture will interact for longer time with evapotranspiration flux than air at higher altitudes and then get enriched with vapor from soil and vegetation.*

*We edited the text as follows:*

*… This positive correlation reflects the imprint of enriched water vapor in the boundary layer moisture. Given the undersaturated conditions during the flights and the typical Mediterranean vegetation of the study area, this enrichment can be attributed to the local evapotranspiration signal. The BLH was then…*

Section 3.4 could be simplified for broader accessibility. Line 391 is particularly confusing. Also, "correlation between standard deviation and vertical flight extent is high…" Does this simply mean that the aircraft measures a wider range of values as it traverses a larger vertical extent?

*Yes, the isotopic variability observed from the aircraft is larger as it traverses a larger vertical extent. We have shortened and rewritten section 3.4 to improve readibility. Here is the modified text (the specific sentence above is reported underlined).*

*Two types of flight patterns were used to investigate the 3D variability of water vapor isotopic signal in detail: vertical profiles (Flights 4-7,11,12, 16) and horizontals scans (8-10,14, 15). Specifically, flights 9 and 10 were designed to investigate the spatial variability at two and three different altitude levels, respectively. Flights 8-10 were performed over the hilly Aubenas area, while flights 14 and 15 were performed over the Rhône Valley, near the town of Montélimar. Fig. 9 focuses on δD, as remote sensing techniques such as LIDAR and satellite instruments only target $H_2^{16}O$ and $HD^{16}O$ absorption bands, not $H_2^{18}O$. $δ^{18}O$ and d-excess maps are provided in Supplementary Material SM4. The $δ^{18}O$, δD, and d-excess variability is discussed hereafter in terms of range (max-min) and of standard deviation (Table 3). The isotopic variability is larger in vertical profiles than in horizontal scans, consistent with expected temperature and humidity gradients. The vertical-to-horizontal range ratio is 2:1 for $δ^{18}O$, δD, and d-excess, while the vertical-to-horizontal standard deviation ratio is 3:1, 4:1, and 1:1, respectively, highlighting δD as the most sensitive parameter in both directions. The standard deviation correlates strongly with the flight extent for vertical flights (0.65, 0.66 and 0.40 for $δ^{18}O$, δD and d-excess, respectively), meaning that a wider range of $δ^{18}O$ δD values were observed as the ULA traversed a larger vertical extent. Horizontal scans show a similar correlation for $δ^{18}O$ and δD with flown-over area, but not for d-excess. Similar to other studies, this dataset also shows a good correlation between the water vapor isotopic composition ($δ^{18}O$ and δD) and the logarithm of the specific humidity, allowing a linear regression model with log(q) as the sole predictor to explain over 90% of δD variability in vertical flights. Notably, for flight 7, a high-altitude sounding, log(q) can explain over 99% of the δD variability. For horizontal scans, the explained variance is smaller but still high on average ($r^2_{δD\ vs\ q}$ = 0.74, see tables in the Supplementary Material SM5 reporting all the r² values). While $COSMO_{iso}$ reproduces observed vertical δ-log(q) patterns, best-fit parameters differ between horizontal and vertical flights in model simulation. Indeed, observed δD vs. log(q) slopes average are very similar 69.4 and 68.9, for vertical and horizontal flights, whereas $COSMO_{iso}$ estimates 65.8 and 122.2, respectively.*

Section 3.5 would benefit from a statement of motivation and significance of results. Also,

Figure 9, while technically interesting, is not particularly intuitive. What does it mean that Flight 12 has larger values than 5, for example? Why is this interesting given where these flights took place?

*To highlight the motivation of this section and the significance of results we included the following statement (beginning of section 3.5) and slightly modified the text as follows:*

*Determining the spatial correlation of water vapor isotopes helps optimize interpolation of sparse observations and assess the ability of CRDS technology to detect fine-scale atmospheric processes using e.g. ULAs. However, given that water vapor isotopic composition is strongly correlated with the specific humidity (and consequently with air temperature), here we explore the variogram of the residuals of the linear model defined between log(q) and δ values.*

*The higher semivariance (the "sill" in the variogram) in flight 12 compared to flight 5 is due to its greater vertical extent, which introduces more variability. These flights reached altitudes of 3200 m and 1700 m, respectively.*

*The key takeaway is that beyond a certain distance (we observe ~1000 m on average for $\delta^{18}O$ and $\delta D$), the isotopic composition of water vapor becomes largely independent of spatial separation, with most of its variability being driven by changes in humidity. We included the above statement when discussing the separation distance (sill and range) in section 3.5*

Fig 10 - I am confused about the axes of this figure. What are the values? Also, is the z scale the same as the x and y scales? It would be nice to get a sense of the size of the structures. Could the x and y axes be shown in units of m or km?

*We are sorry for the unusual and confusing metrics. The axes values were already meters but for a UTM projection coordinates system (hence distance from a meridian and from the equator). We plotted the data again by centering the coordinate system over the flight area, as shown below. We also included geographic north as a reference and reported the vertical exaggeration (~9) in the caption. The same was applied to Fig. 11.*

[Figure]

L 461 - "At L1200, close to the boundary layer height, the r2 significantly increases (0.83) and spatial features in the residual field are more evident (Fig. 10b)." I feel as though I've missed something here. If the r2 is high, meaning that the model predicts dD well, why are the residual features more substantial? It would help to provide a bit more physical context for what the results mean. Later, L 488 states: "the more evident the spatial features in the residual fields are, the smaller the r2." Are these contradictory statements?

*Yes, the $r^2$ is high when the regression model predics dD well: smaller residuals → isotopic variaiblity mostly driven by changes in humidity. We believe the residual features are more substantial at L1200 because they are spatially clustered and non randomly scattered over the scan area. We speculate that the slightly larger δD variability at L1200 (3‰ vs 1‰ for lower scans) can be attributed to exchange of water vapor with different isotopic signatures between the boundary layer and the free atmosphere.*

*We highlighted this aspect by adding some details about the magnitude of the residuals and editing the text accordingly:*

*At L1200, close to the boundary layer height, the $r^2$ significantly increases (0.83) and the spatial features in the residual field are more evident (Fig. 10b). While the q variability remains similar across levels (~0.1 g kg$^{-1}$), the slightly larger δD variability at L1200 (3‰ vs 1‰) can be attributed to short-ranged exchange of water vapor with different isotopic signatures between the boundary layer and the free atmosphere….*

*[]….Therefore, we speculate that the ULA may have captured intermittent coherent structures which are commonly observed at the boundary layer top over terrain with high surface roughness (Thomas and Foken, 2007) while residual fields for horizontal scans*

*within the lowermost layers are mostly driven by instrumental uncertainty (~1‰ for $\delta$D).*

*About the sentence at L488, thank you for pointing out such statement, which can be confusing. We removed the sentece accordingly. The key message we want to highlight here is that Flight 15 is representative of a well mixed boundary layer (midday-afternoon) where $\delta$D is strongly correlated to changes in humidity, while Flight 14 is representative of boundary layer development at early stage (morning), with evident spatial clustering of model residuals. We included the following statement at the end of section 3.7:*

*In summary, the variability in the residual field is linked to early-stage boundary layer development during flight 14, while for flight 15, it reflects a well-mixed boundary layer state.*
* * *
L 494 - "Having seen that water vapor mixing ratio can provide a first-order approximation of the vertical and horizontal water vapor isotopic structure in the atmosphere." To what extent does this depend on measuring in a fairly well-mixed boundary layer in the midlatitudes in September? Also, where are the model formulae stated?

*We observed a well-mixed boundary layer in most flights, so our findings may not be directly applicable to other case studies. To strengthen our analysis, we adopted an exploratory approach, integrating our observations with other airborne datasets representing various climatic zones and weather situations. We have emphasized this point further in the conclusions:*

*… Although our observations cover a short period and a limited geographical area, combining our dataset with other airborne measurements allowed us to approximate full-column $\delta$D as a function of specific humidity gradient. This, in turn, improves the scaling of surface $\delta$D observations to the tropospheric column, enhancing $\delta$D satellite validation.*

*About the model formulas: it is not clear if the referee is mentioning the mixing model/rayleigh model or to the log-linear model we fitted on our observations + extra datasets.*

*For the former, we refer to the equations in Noone (2012), as stated in the methods sections 2.7*

*Specifically, here we report only the principal assumptions behind the two approaches, and we refer to equations in Noone (2012) for both models.*

*For the latter, the formula is equation (2) .*
* * *
Figure 12 - I worry that an RMSE of 20 permil in dD is about the same order of magnitude as the actual signal of isotope ratio change in the boundary layer (see Figure 6c). Also, d-excess is essentially constant in the boundary layer, and thus the prediction range is quite small. Knowing this, how meaningful are the simple models from whence the errors are derived?

*We agree that it is actually a large error for vertically resolved data but it could be useful for computing the total column values with a simple log model (as we show in section 4.3). Indeed, considering our study and other datasets, the mean difference between observed and modelled (mixing model) weighted average δD is 4.2 ± 12.7 ‰ as shown in Fig. 15.*

L 560 - "Our results hence suggest that water vapor isotopes could be used as a proxy for studying boundary-layer development." This assertion raises the question: why not just use water vapor, which can be measured at much higher temporal resolution? Would studying water vapor structures have given a different result?

*Studying humidity alone would likely yield different results. Here, we report the case of flight 14, but using q as the variable of interest, instead of log(q)-δD model residuals. While spatial patterns are still visible, they appear to be on a different (larger) scale. This aligns with our analysis of separation distances for humidity, which predicts a drop in spatial correlation at scales of ~100 km. In this context, stable isotopes seem to provide additional information at finer scales. Our findings are also consistent with other modeling approaches—for example, OSSE simulations have shown that incorporating isotopic information into water vapor concentrations improves weather forecasts (Yoshimura et al., 2015).*

[Figure]

L 582 - "a few data points within the boundary layer can be used to estimate the vertical profile of the water vapor isotopic composition". But given the errors in Figure 12, could one simply assume a constant BL isotope ratio and obtain errors of equivalent magnitude? In that case, wouldn't one data point suffice?

*The isotopic ratio will likely change with altitude because of change of temperature (and humidity). In principle, yes, one point would suffice if that point can be representative of the boundary layer (e.g. not too close to the ground to minimize the impact from surface*

*flux) but scaling with humidity profile will be required.*

L 591 - The paper assumes a transpiration contribution. Couldn't COSMOiso verify this? Evaluating such a hypothesis seems like an appropriate application for the comprehensive simulations.

*Unfortunately it is not possible to verify this directly with the model data because COSMOiso provides only the total surface moisture flux, without partitioning between bare soil evaporation, transpiration etc.*

*On this topic, also Referee #2 suggested that our approach with direct comparison with GNIP precipication might be not representative of the flux composition, while water vapor in isotopic equilibrium should provide a better estimates of the evapotranspiration isotopic signal in the study area. To better investigate the observed shift in flux composition, we approached the issue in two steps (the same answer is attached on Reply to Referee #2 file).*

***First**, we estimated the isotopic composition of water vapor in equilibrium with precipitation using GNIP data from the Avignon station (~100 km south of the study area) and monthly mean air temperature data from Avignon (ECA&D) for the period 1997–2021. The GNIP values were corrected for the altitude difference ($\Delta z$ = 255 m) using lapse rates of 0.2‰ for $\delta^{18}O$ and 1‰ for $\delta D$ (Masiol et al., 2021).*

***Table**: Isotopic composition of water vapor in equilibrium with precipitation at Avignon between 1997 and 2021, corrected for altitude effect. The *starred values are for the temporal interval of the field campagin.*

| Date | T (K) | d$^{18}$O Eq.Vap | dD Eq. Vap |
|---|---|---|---|
| Sep-97 | 293,4 | -13,38 | -105,4 |
| Sep-98 | 292,0 | -13,92 | -109,9 |
| Sep-99 | 294,2 | -15,07 | -116,9 |
| Sep-00 | 293,2 | -15,64 | -122,1 |
| Sep-01 | 290,8 | -15,94 | -125,6 |
| Sep-02 | 291,1 | -16,41 | -124,4 |
| Sep-03 | 292,2 | -13,89 | -107,6 |
| Sep-04 | 293,2 | -17,17 | -134,0 |
| Sep-05 | 292,5 | -16,78 | -130,0 |
| Sep-06 | 294,1 | -14,11 | -107,0 |
| Sep-07 | 292,0 | -18,04 | -135,9 |
| Sep-08 | 291,3 | -14,51 | -110,8 |
| Sep-09 | 293,2 | -21,38 | -163,9 |
| Sep-10 | 291,7 | -15,18 | -115,5 |
| Sep-11 | 293,9 | -14,78 | -112,9 |
| Sep-12 | 293,0 | -15,49 | -119,4 |
| Sep-13 | 292,8 | -14,70 | -112,3 |
| Sep-14 | 293,7 | -15,32 | -118,1 |
| Sep-15 | 292,0 | -12,34 | -97,6 |

|        |        |         |         |
|--------|--------|---------|---------|
| Sep-16 | 294,8  | -14,25  | -108,1  |
| Sep-17 | 291,1  | -11,97  | -103,4  |
| Sep-18 | 294,3  | -11,89  | -95,0   |
| Sep-19 | 294,1  | -14,65  | -117,0  |
| Sep-20 | 293,6  | -17,04  | -131,0  |
| Sep-21* | 293,9* | -13,38* | -107,5* |
|        |        |         |         |
|        | **Average** | -15,09 | -117,3 |
|        | **Std. Dev.** | 2,05 | 14,5 |

*The analysis of water vapor in equilibrium with precipitation suggests that $\delta^{18}O_F$ aligns more closely with water vapor in equilibrium with precipitation, which is at first order comparable to evapotranspiration signal over the study area (and also consistent wth our $\delta^{18}O$ vs $\delta D$ analysis in section 3.3). As the Referee #2 correctly pointed out, this approach is not entirely precise due to the assumption of saturation, but it provides a more reliable comparison than directly linking water vapor and precipitation. Please note that we have now computed the $\delta^{18}O$ flux composition (estimated as the keeling-plot intercept) using the 150m vertically binned flight observation data. The results do not change significantly but we have specified how we have calculated the flux for completeness.*

***Second**, as suggested by Referee #1 we delved more into the model data to verify if it is possible to separate the transpiration to evapotranspiration signal. Unfortunately, only the total surface moisture flux, and its isotopic composition, was stored in the COSMOiso output data, without partitioning between bare soil evaporation, transpiration etc. Hence, we analysed the variability of surface moisture flux throughout the day in COSMOiso data. On 18 September, $\delta^{18}O_{F-COSMOiso}$ interpolated along flight tracks f04 to f07 changed from –3.13 to –5.15‰, indicating a shift in the flux composition also in the model. A clear diurnal cycle in $\delta^{18}O$ _{F-COSMOiso} can be observed on 21 and 22 September at the model grid point corresponding to the study site, as shown in the following figure:*

[Figure]

***Figure**: Isotopic composition of the total surface moisture flux ($\delta^{18}O_{F-COSMOiso}$) at the Lanas Airfield model grid point for 21 and 22 September 2021. Red-highlighted dots indicate the times of the flights on these specific days.*

*Thus, both observations (Keeling plot intercepts representative of $\delta^{18}O_F$) and model simulations ($\delta^{18}O$ ~F-COSMOiso~) show a daytime shift in the isotopic composition of the flux. We therefore maintain our conclusion that $\delta^{18}O_F$ varies throughout the day. While we do not claim whether evaporation or transpiration dominates this shift, we emphasize that assuming turbulent mixing either one endmember is changing its isotopic signature or multiple endmembers contribute to the boundary layer moisture composition.*

*We edited the text as follows:*

*For example, we estimated a change from $\delta^{18}O_F$ = -6.12‰ at 5 UTC to $\delta^{18}O_F$ =-13.38‰ at 15 UTC on 18 Sep (flights 4 to 7) with keeling-plot method applied on 150 m binned vertical profiles. Intriguing, the average $\delta^{18}O$ of water vapor in isotopic equilibrium with precipitation for September 2021, estimated from altitude-corrected GNIP (IAEA) data and air temperature records from Avignon (~100 km south, ECA&D) is −13.38‰. Although this estimate assumes saturation and equilibrium, making it approximate, it supports the hypothesis that evapotranspiration influences boundary layer moisture during the day. However, the observed shift in the $\delta^{18}O_F$ end-member composition from morning to afternoon also indicates that assigning a constant isotopic signature based on nearby precipitation is not reliable.*

---

## Author Comment (AC2)

Reply to

RC1: 'Comment on egusphere-2024-3394', Anonymous Referee #2, 28 Jan 2025

In the following text the Referee's (#2) comments are reported as normal lead text and the authors answers are reported in grey shaded box like this one *with italic font*. The edited text in the manuscript is in *red color*.

**General thoughts**

Zannoni et al. presents a new dataset of atmospheric water-vapor isotopes and dexcess above southern France. They achieve this by integrating a Picarro instrument into an ultralight aircraft (ULA) and performing a series of flights of differing flight patterns, both horizontally and vertically. They pair these observations with model realizations of water-vapor isotope composition via COSMOiso to evaluate claims they make. The authors describe several key conclusions which are supported by both their observations and their model results. They find that (i) vertical mixing is a key determinant of atmospheric isotope composition, (ii) the bottom mixing endmember is likely evapotranspiration and (iii) fine-scale structure exists that aren't captured by models. This study works towards understanding the representation of water-isotope composition in the atmosphere with the framing of both understanding basic science but only serving alternative measurement techniques such as remote sensing and groundbased observation systems.

This study is pioneering in several ways. For observations, the approach is a logistically-light implementation of in-situ vapor measurement including a rigorous calibration scheme. With those observations, they apply two different frameworks to explain vertical isotope distribution, a Rayliegh framework, and a vertical mixing framework. Within error bars, they are unable to distinguish between which framework might be the best fit, but with additional context clues, such as the consideration of the bottom endmember, they ascribe the atmospheric column to be best described by mixing.

*We are grateful to Referee #2 for her/his valuable comments and insightful suggestions, which have helped improve our manuscript. We believe we have addressed all the Referee's comments. Below, our detailed step-by-step responses.*

*Please note that we have slightly trimmed the methodological section. The text was edited specifically at the following locations (line numbers refer to the original preprint):*

***Lines 122-126***: *moved to Supplementary Material SM0. Main text in the manuscript edited as follows:*
*The reader is referred to Supplementary Material SM0 for details on frequency of usage, values of isotope standards and calibration performances of the CRDS analyzer.*

***Lines 197-205***: *Rephrased in a more synthetic way*

In the horizontal domain, with the aid of the isotope-enabled model COSMO$_{iso}$, the authors find the distance before air parcels can be considered statistically unrelated.

The authors don't report this as a major conclusion, but I disagree with that approach. It most certainly informs the design of a measurement campaign that would work from another conclusion of theirs, that a surface vapor measurement may inform the total column composition in a log relationship.

*Thank you for highlight this point. We have modified the key findings in the abstract to put more emphasis on this aspect. We have also slightly edited the first and second key finding in the abstract and conclusion to put more emphaisis on the altitude we are referring to (lowermost boundary layers vs lower troposphere). Here is the edited sentense in the abstract:*

*The key findings of this study are that (i) at hourly and sub-daily scales, vertical mixing is the primary driver of isotopic variability in the lowermost troposphere above the study site, and (ii) evapotranspiration significantly impacts the water vapor isotopic signature, specifically in the boundary layer, as revealed by the $\delta^{18}O$-$\delta D$ relationship; (iii) while water vapor isotopes generally follow large-scale humidity patterns with separation distances that might range up to 100–300 km, they also reveal distinct small-scale structures (~100 m) that are not fully explained by humidity variations alone, highlighting sensitivity to additional fine-scale processes.*

The authors also find fine-scale microstructures on the order of 100s of meters. Observational evidence of these microstructures is new and a relevant contribution to the broader body of work on watervapor isotopes. They attribute this to potentially being from the surface terrain's aspect, producing thermals that affect isotope composition aloft. This may or may not be true, but the observation of microstructures at this scale provides a landscape for similar testable hypothesis that are a benefit to the community.

*Regarding the discussion on such fine-scale structure, following also more clarifications requested by Referee #1, we have expanded section 4.2 by estimating the characteristic size and lifetime of such structures (with references to previous studies), and their significance (along with their limitations with CRDS technology) for studying boundary layer development and turbulent mixing processes.*

The paper is mostly clear and well-written. With its expanded and detailed approach to observational methods, this paper should encourage more measurement of its kind. I have several comments for the text which do not disagree with their main conclusions but work to better represent them. I additionally have several technical edits to text. I hope this type of research continues and I support the statement that Zannoni et al. contributes a valuable study to the water-vapor isotope field in the scope of this journal.

*We are grateful to Referee #2 for the constructive approach in this review. We have addressed all the technical edits, as reported hereafter.*

Line 149: The author's note after vehicle vibration calibration:
"Note that shocks and vibrations are expected to be less pronounced when the ULA is airborne, thus we provide here a conservative estimate of the vibrational impacts." This is factually inaccurate. Running the engine on the ground does not adequately probe all vibrational modes, some of which will be shared by the Picarro's cavity. From my experience in aviation, I understand that official advice on the topic of aircraft vibrations

for airworthiness is to test the aircraft in all operating conditions. I have attached the American Federal Aviation Administration's Advisory Circular on the topic still in use in aircraft manufacturing. See Chapter 1: Section 2: General Considerations: a (1.2.a). Still, calibration for vibration at all is an improvement on previous work in Chazette et al 2021 and I believe constitutes current due diligence on the subject. My recommendation is to remove the sentence claiming conservative estimates in favor of an acknowledgement that not all vibrational modes can be reasonably tested in the scope of the study.

*Thank you for pointing out this weakness on the evaluation of the uncertainty and thank you for the useful reference. We have specified that our attempt may be representative of vibrations only during the stand-by phase of the aricraft on the ground. We edited the text accordingly:*

*These ADEV values can therefore be assumed representative of the instrumental precision at 1 second averaging time and at $q = 8.2$ g kg$^{-1}$ in the Taxi to Runaway phase. On this latter point, it is worth to be noted that our approach does not adequately probe all vibrational modes, hence instrumental precision might be worse. Indeed, instrument performances should be evaluated under all normal operating conditions to obtain the full spectrum of vibrational noise (AC No. 20-66, 1970).*

Line 255: The author's would benefit from describing the time step of COSMOiso. Using ECHAM6-wiso as a boundary condition at 6-hour resolution, it appears that COSMOiso might be the same which would affect the interpretability of comparisons to observations. It seems that the time step might be 1 hour based on a careful reading of Villiger el al. 2023 but either way, this should be clear in this text.

*We have now provided the information about the model time step (30s and 20s for the 10km and 2km simulations, respectively) and the temporal resolution of COSMO$_{iso}$ (1 hour).*

*..., and with a model time step of 30s for the 10km and 20s for 2km simulation. The COSMOiso fields are output at a 1-hourly resolution.*

Figure 5: The days selected to plot does not follow a pattern I can recognize and all days are discussed in the text. In fact, if looking to plot days with the most flights, the plotted days are ill-suited to represent measurements. Consider days with high number of flights, or including a figure in supplementary materials with all flight days.

*Figure 5 serves as an indication on the general weather situation during the campaign. Specifically, for the development of sparse rain on 18-19 and 20-21 Sep which interested the study area, but not directly the flight operational conditions.*

Line 323: The author's make a statement on observations of d-excess:
"Among the flights which reached altitudes > 3000 m (flights 3, 7, 11-16), only flight 7 exhibits a consistent positive deviation of d-excess from the mean value observed at lower altitudes, ranging from 12 ± 2 ‰ at 2000 m to 19 ± 3‰ at 3000m."

And then follow with the statement:
"The d-excess increase as a function of the altitude is a well-known feature of atmospheric water vapor and typical of clear sky conditions".

From my understanding of the text, the authors note lack of an increase of d-excess with altitude for many flight yet immediately state that the opposite is typical. If my understanding on their intent is correct, I would expect that the claim is cited and some justification given for why the result deviates from the norm.

*We expected an increase in d-excess with altitude, as commonly reported in airborne measurements. However, this trend was not observed in our data, except for flight 7, which also showed a marked humidity decrease near 3000 m. In our case, the absence of a significant d-excess increase may be explained by a well-mixed boundary layer and relatively uniform RH profiles. We have clarified the statement noted by the Referee, emphasizing the role of low humidity at high altitudes in enhancing d-excess through non-equilibrium fractionation between water vapor, liquid water, and ice. To support this interpretation, we reference other high-altitude studies, including modeling approaches.*

*....We speculate that the absence of a similar trend in the other flights may be due to a well-mixed boundary layer and relatively homogeneous RH profiles. Notably, the d-excess increase during flight 7 begins after passing a relative humidity maximum around 1800–2000 m, which may correspond to the cloud base and suggest the impact of cloud droplets evaporation. The d-excess increase as a function of the altitude is a well-known feature of atmospheric water vapor, typically resulting from non-equilibrium fractionation processes under low humidity at higher elevations, as shown by both in situ observations and model studies (e.g. Bony et al., 2008; Samuels-Crow et al., 2014).*

*Bony, S., C. Risi, and F. Vimeux. Influence of convective processes on isotopic composition of precipitation ($\delta^{18}O$ and $\delta D$) of precipitation and watervapor in the tropics: 1. Radiative-convective equilibrium and Tropical Ocean-Global Atmosphere-Coupled Ocean-Atmosphere Response Experiment (TOGA-COARE) simulations, J. Geophys. Res., 113, D19305, https://doi.org/10.1029/2008JD009942, 2008.*

*Samuels-Crow, K. E., J. Galewsky, Z. D. Sharp, and K. J. Dennis: Deuterium excess in subtropical free troposphere water vapor: Continuous measurements from the Chajnantor Plateau, northern Chile, Geophys. Res. Lett., 41, 8652–8659, https://doi.org/10.1002/2014GL062302, 2014.*

Line 349: The author's state "Discrepancies between observed and modelled d-excess can be attributed to differences in simulated and observed $\delta^{18}O$ and $\delta D$ at high altitude,…". This statement is definitional of dxs and can be removed in favor of focusing on the rest of the sentence.

*Done.*

Figure 11: The author's reference that terrain aspect might be the cause for microstructure in water vapor isotopes aloft. However, figure 11 poorly shows this. I suggest rearranging such that this claim seems more plausible and include a color bar for the grey elevation at the bottom

*Following Referee #2 suggestion we have included a second gray-scale bar, which reports*

*the altitude scale, hence highlighting where the Rhône Valley and the nearby slopes are located. We believe that this information, together with contour lines of terrain elevation, helps the reader identifiyng where the slopes are more steep, and where thermals are more likely to develop.*

*Please also note that following the Referee #1 suggestion for Figure 10 and 11, we centered the horizontal coordinate sytem on the flown-over area. The axis values are in meters, based on the WGS84 UTM Zone 31 projection, but distances can now be more easily evaluated visually. We now also report in the Figures' caption the vertical exhageration values (to enhance visualization of vertical features).*

Line 588: The authors note that near surface vapor is similar the recent GNIP precipitation data. I believe this isn't the comparison to make for evapotranspiration. The better one would be to compare vapor data to water vapor in equilibrium with GNIP precipitation. Of course, there is an open variable here in surface humidity but even a blind choice of being at saturation would be a better comparison target for comparing the surface end member in a mixing model.

*Thank you for highlighting this important aspect of our analysis. We agree that the precipitation signal alone may not fully represent the isotopic composition of the evapotranspiration flux, since evapotranspiration includes both soil evaporation (which introduces isotopic fractionation) and transpiration (which does not). Following Referee #2's suggestion and Referee #1's remarks on the role of transpiration, we further investigated this issue.*

***First***, *we estimated the isotopic composition of water vapor in equilibrium with precipitation using GNIP data from the Avignon station (~100 km south of the study area) and monthly mean air temperature data from Avignon (ECA&D) for the period 1997–2021.* *The GNIP values were corrected for the altitude difference* *(Δz = 255 m) using lapse rates of 0.2‰ for δ$^{18}$O and 1‰ for δD (Masiol et al., 2021).*

***Table***: *Isotopic composition of water vapor in equilibrium with precipitation at Avignon between 1997 and 2021, corrected for altitude effect. The \*starred values are for the temporal interval of the field campagin.*

| Date | T (K) | δ$^{18}$O Eq.Vap | δD Eq. Vap |
|---|---|---|---|
| Sep-97 | 293,4 | -13,38 | -105,4 |
| Sep-98 | 292,0 | -13,92 | -109,9 |
| Sep-99 | 294,2 | -15,07 | -116,9 |
| Sep-00 | 293,2 | -15,64 | -122,1 |
| Sep-01 | 290,8 | -15,94 | -125,6 |
| Sep-02 | 291,1 | -16,41 | -124,4 |
| Sep-03 | 292,2 | -13,89 | -107,6 |
| Sep-04 | 293,2 | -17,17 | -134,0 |
| Sep-05 | 292,5 | -16,78 | -130,0 |
| Sep-06 | 294,1 | -14,11 | -107,0 |
| Sep-07 | 292,0 | -18,04 | -135,9 |
| Sep-08 | 291,3 | -14,51 | -110,8 |

| | | | |
|---|---|---|---|
| Sep-09 | 293,2 | -21,38 | -163,9 |
| Sep-10 | 291,7 | -15,18 | -115,5 |
| Sep-11 | 293,9 | -14,78 | -112,9 |
| Sep-12 | 293,0 | -15,49 | -119,4 |
| Sep-13 | 292,8 | -14,70 | -112,3 |
| Sep-14 | 293,7 | -15,32 | -118,1 |
| Sep-15 | 292,0 | -12,34 | -97,6 |
| Sep-16 | 294,8 | -14,25 | -108,1 |
| Sep-17 | 291,1 | -11,97 | -103,4 |
| Sep-18 | 294,3 | -11,89 | -95,0 |
| Sep-19 | 294,1 | -14,65 | -117,0 |
| Sep-20 | 293,6 | -17,04 | -131,0 |
| Sep-21* | 293,9* | -13,38* | -107,5* |
| | | | |
| | **Average** | -15,09 | -117,3 |
| | **Std. Dev.** | 2,05 | 14,5 |

*The analysis of water vapor in equilibrium with precipitation suggests that $\delta^{18}O_F$ aligns more closely with water vapor in equilibrium with precipitation, which is at first order comparable to evapotranspiration signal over the study area (and also consistent wth our $\delta^{18}O$ vs $\delta D$ analysis in section 3.3). As the referee correctly pointed out, this approach is not entirely precise due to the assumption of saturation, but it provides a more reliable comparison than directly linking water vapor and precipitation. Please note that we have now computed the $\delta^{18}O$ flux composition (estimated as the keeling-plot intercept) using the 150m vertically binned flight observation data. The results do not change significantly but we have specified how we have calculated the flux for completeness.*

**Second**, *as suggested by Referee #1 we delved more into the model data to verify if it is possible to separate the transpiration to evapotranspiration signal. Unfortunately, only the total surface moisture flux, and its isotopic composition, was stored in the COSMOiso output data, without partitioning between bare soil evaporation, transpiration etc. Hence, we analysed the variability of surface moisture flux throughout the day in COSMOiso data. On 18 September, $\delta^{18}O_{F-COSMOiso}$ interpolated along flight tracks f04 to f07 changed from –3.13 to –5.15‰, indicating a shift in the flux composition also in the model. A clear diurnal cycle in $\delta^{18}O_{F-COSMOiso}$ can be observed on 21 and 22 September at the model grid point corresponding to the study site, as shown in the following figure:*

[Figure]

***Figure***: *Isotopic composition of the total surface moisture flux ($\delta^{18}O_{F-COSMOiso}$) at the Lanas Airfield model grid point for 21 and 22 September 2021. Red-highlighted dots indicate the times of the flights on these specific days.*

*Thus, both observations (Keeling plot intercepts representative of $\delta^{18}O_F$) and model simulations ($\delta^{18}O_{F-COSMOiso}$) show a daytime shift in the isotopic composition of the flux. We therefore maintain our conclusion that $\delta^{18}O_F$ varies throughout the day. While we do not claim whether evaporation or transpiration dominates this shift, we emphasize that assuming turbulent mixing either one endmember is changing its isotopic signature or multiple endmembers contribute to the boundary layer moisture composition.*

*We edited the text as follows:*

*For example, we estimated a change from $\delta^{18}O_F$ = -6.12‰ at 5 UTC to $\delta^{18}O_F$ =-13.38‰ at 15 UTC on 18 Sep (flights 4 to 7) with keeling-plot method applied on 150 m binned vertical profiles. Intriguing, the average $\delta^{18}O$ of water vapor in isotopic equilibrium with precipitation for September 2021, estimated from altitude-corrected GNIP (IAEA) data and air temperature records from Avignon (~100 km south, ECA&D) is –13.38‰. Although this estimate assumes saturation and equilibrium, making it approximate, it supports the hypothesis that evapotranspiration influences boundary layer moisture during the day. However, the observed shift in the $\delta^{18}O_F$ end-member composition from morning to afternoon also indicates that assigning a constant isotopic signature based on nearby precipitation is not reliable.*

Technical corrections
Line 366: Boundary layer height, referred as "blh" should be capitalized as an acronym

*Corrected here and elsewhere in the manuscript.*

Line 666: "spati 6al anisotropy" is a typo

*Corrected.*